# Extracellular release of two peptidases dominates generation of the trypanosome quorum-sensing signal

Mabel Deladem Tettey [1], Federico Rojas[1,2 ✉] & Keith R. Matthews [1 ✉]

Trypanosomes causing African sleeping sickness use quorum-sensing (QS) to generate transmission-competent stumpy forms in mammalian hosts. This density-dependent process is signalled by oligopeptides that stimulate the signal transduction pathway leading to stumpy formation. Here, using mass spectrometry analysis, we identify peptidases released by trypanosomes and, for 12 peptidases, confirm their extracellular delivery. Thereafter, we determine the contribution of each peptidase to QS signal production using systematic inducible overexpression in vivo, and confirm this activity operates through the physiological QS signalling pathway. Gene knockout of the QS-active peptidases identifies two enzymes, oligopeptidase B and metallocarboxypeptidase 1, that significantly reduce QS when ablated individually. Further, combinatorial gene knockout of both peptidases confirms their dominance in the generation of the QS signal, with peptidase release of oligopeptidase B mediated via an unconventional protein secretion pathway. This work identifies how the QS signal driving trypanosome virulence and transmission is generated in mammalian hosts.

[1] Institute for Immunology and Infection Research, School of Biological Sciences, University of Edinburgh, Edinburgh EH93FL, UK. [2] Present address: Biomedical and Life Sciences, Faculty of Health and Medicine, Lancaster University, Lancaster LA1 4AT, UK. ✉email: federico.rojas@lancaster.ac.uk; keith.matthews@ed.ac.uk

African trypanosomes are responsible for the human disease Human African trypanosomiasis (sleeping sickness) as well as the important livestock disease Animal African trypanosomiasis (nagana)[1,2]. These diseases affect health and wealth in afflicted regions of sub-Saharan Africa, infecting thousands of people each year and causing loss to agriculture in the region of $4 billion annually, this acting as a significant driver of poverty. Trypanosomes are transmitted by tsetse flies, with parasite development in the mammalian host contributing to transmission competence[3]. Specifically, proliferative slender form parasites differentiate from non-proliferative stumpy form parasites as a preadaptation for tsetse uptake, these stumpy forms exhibiting, for example, mitochondrial activity in preparation for the nutritional environment of the arthropod gut[4,5]. The development from slender to stumpy forms is predominantly a density-dependent phenomenon such that slender forms multiply to sustain the infection whereas arrested stumpy forms dominate at the peak of the infection and throughout the chronic phase of the infection[6]. This, in combination with the parasites' sophisticated immune evasion strategy, antigenic variation, generates the undulating infection dynamic characteristic of trypanosome infections.

The trypanosome quorum sensing (QS) response is understood in some molecular detail. Firstly, components of the intracellular molecular cascade that transduces the QS signal have been identified by a genome-wide screen selecting for parasites unable to respond to a QS signal mimic after their silencing by RNA interference[7]. This identified several regulators including protein kinases, phosphatases and RNA binding proteins, reflecting the predominant control of trypanosome gene expression at the post-transcriptional level[8]. At the cell surface, a multimembrane spanning protein TbGPR89 has also been discovered that contributes to QS signalling[9] and this molecule has oligopeptide transport capability. The identification of this molecule highlighted the potential for oligopeptides to act as the external signal driving QS and, indeed, complex oligopeptide mixtures precipitate slender to stumpy differentiation, with synthetic oligopeptides demonstrating specificity of the response.

The discovery that short peptides could promote trypanosome differentiation raised the potential that parasite-derived peptidases could generate the extracellular signal for QS. Indeed at least two parasite-derived proteases have been identified that are released by bloodstream form parasites and retain activity in the serum of infected animals[10,11]. These are prolyl oligopeptidase and pyroglutamyl peptidase, and each of these peptidases promotes QS when ectopically over expressed in transgenic bloodstream form parasites[9]. In consequence, a model has been developed whereby parasite-derived peptidases generate the QS oligopeptide signal, with their abundance and resultant peptide-generating activity acting as a proxy for parasite density and hence stumpy formation[12]. Consistent with this model, basement membrane extracts support efficient parasite differentiation in culture, the enriched extracellular matrix components acting as likely substrates of released proteases[13,14].

A systematic functional screening of trypanosome proteases has been carried out using targeted RNA interference against each of 30 peptidases encoded in the trypanosome genome[15]. This revealed that only 1 peptidase produced by the parasites was essential, a signal processing peptidase. Other peptidases were apparently dispensable, at least via RNAi, and their depletion did not cause growth effects in transgenic parasites. However, studies were largely confined to the analysis of parasites grown in culture and the cellular distribution or capacity for release for the targeted peptidases was not analysed, although proteases have been detected in the excretory/secretory products of *Trypanosoma congolense* and *Trypanosoma gambiense*[16]. Overall, therefore, the contribution of parasite-derived peptidases to QS is clear, but the enzymes and activities that contribute to driving the process in vivo have not been determined.

Here we analyse the peptidases released by trypanosomes at different stages of development in the mammalian host and during onward differentiation to tsetse midgut forms. The released peptidases are then evaluated for their ability to contribute to the generation of the QS response, signalled through the identified molecular signalling pathway. In common with an increasing number of parasite-released proteins[17], we find their release occurs through a pathway distinct from classical secretion. In combination, this identifies that two major peptidases dominate the trypanosome QS signalling response, these acting in concert, likely with minor additional contribution from other peptidases, to drive the parasite QS signal generation.

## Results

**The released proteome of bloodstream trypanosomes.** In order to identify proteins released by trypanosome parasites in the mammalian host, populations of slender or stumpy parasites were derived from murine infections (stumpy forms) or culture (slender forms; necessary to derive sufficient numbers for analysis), purified and then incubated in serum-free Creek's Medium for 2 h. Also, to include proteins released by parasites upon the early events of transmission to procyclic forms, stumpy form parasites were incubated in HMI-9 medium containing 6 mM cis aconitate and then, after 1 or 4 h, parasites were washed and incubated in serum-free Creek's medium containing cis-aconitate for 2 h (Fig. 1a). This provided information on released molecules in bloodstream forms and during the synchronous differentiation of stumpy forms to procyclic forms, after a total of 3 h or 6 h exposure to the differentiation signal cis-aconitate. After cells were separated by centrifugation, the cell pellet and culture medium derived material was analysed by western blotting with antibody to EF1-alpha to ensure the absence of contamination of the supernatant with cytosolic protein from cell lysis (Supplementary Fig. 1a–d). For all samples, three biological replicates were analysed by mass spectrometry, with very good reproducibility between the replicates (slender; >0.947; stumpy >0.769; 3 h differentiation >0.942; 6 h differentiation >0.890; Fig. 1b). We selected proteomic identifications based on protein detection in at least two out of three biological replicates and with at least two distinct peptides identified from the parent protein (Supplementary data 1). Comparing the released proteome of slender and stumpy forms identified 56 proteins common between the two developmental forms, plus 20 that were detected only in slender forms and 29 that were only detected in stumpy forms. Analysis of the differentiating cells by the same criteria revealed 92 that were specifically detected after 3 h in cis-aconitate, and 52 exclusively detected at 6 h, whereas 83 proteins were common between the two differentiation-enriched released proteomes. 19 proteins were common between all of the datasets (i.e., slender, stumpy, 3 h and 6 h differentiation) (Supplementary data 2).

Among the released molecules identified from all the samples, many peptidases were detected (Supplementary Fig. 1e; Fig. 1c) and these were of particular interest given the potential contribution of this group of molecules to the parasites' QS response[9] and previous evidence of peptidase release during the course of trypanosome infections[10,11]. In total, 12 peptidases were detected in the overall dataset of released proteins (Supplementary Fig. 1e). In a previous global screen for protease-related phenotypes[15], none of the identified peptidases was detected to have any effect on parasite growth in monomorphic bloodstream form cells in culture, at least as detected by RNA interference. However, monomorphic cells are unable to report on developmental phenotypes in the mammalian

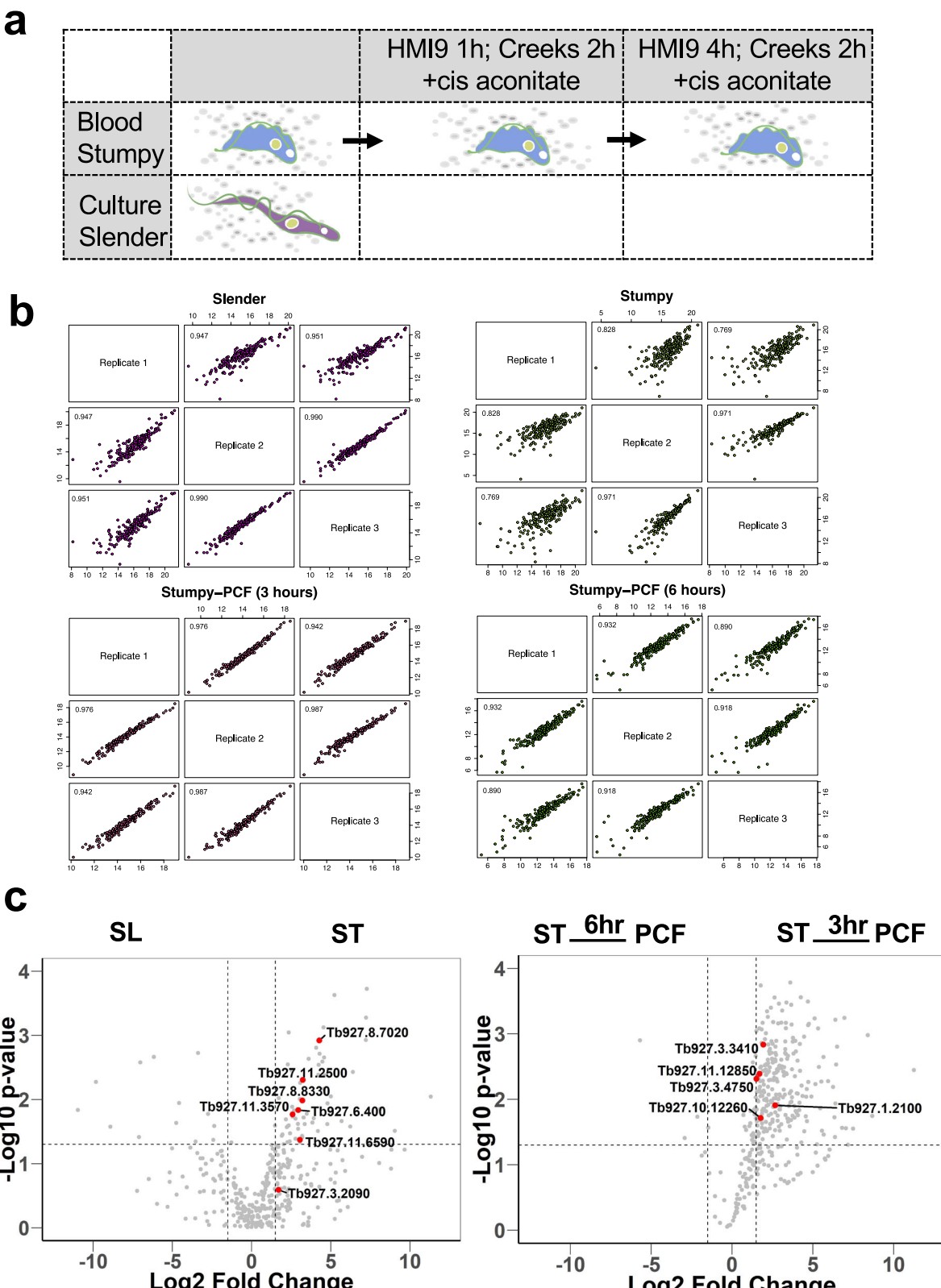

**Fig. 1 Mass spectrometry analysis of released trypanosome proteins. a** Schematic representation of the samples used for the analysis of the secreted proteome from bloodstream slender forms, stumpy forms or stumpy form parasites undergoing differentiation to procyclic forms. **b** Scatter plots showing the reproducibility of three biological replicates of secreted material released from slender forms, stumpy forms or cells 3 h or 6 h into differentiation to procyclic forms. **c** Volcano plots of proteins detected as released from either slender or stumpy forms (left-hand panel) or during early differentiation (right-hand panel) highlighting released peptidases. Log2 of the fold change (FC) is plotted on the *x*-axis and the −log10 of the *p* value (ANOVA) on the *y*-axis. The vertical dotted lines indicate a fold change >1.5 and the horizontal dotted line a *p* value < 0.05. Red dots represent peptidases detected in the dataset; significance thresholds are indicated by dotted lines. SL is slender, ST is stumpy and PCF is procyclic. Source data are provided as a source data file.

host and therefore we specifically explored the released molecules from pleomorphic bloodstream forms further.

**The detected 12 peptidases are released by bloodstream parasites.** Initially, we sought confirmation that the peptidases detected in the released proteome were indeed derived from intact trypanosomes rather than from damaged or lysed parasites. In the absence of available antibodies to detect each peptidase, we tagged one allele for each peptidase at its endogenous genomic locus, using a Ty1 epitope tag sequence, generating an N-terminal 10 amino acid tag on the encoded protein detectable with the Ty1-specific BB2 antibody[18]. The resulting cell lines were assayed for release of the tagged peptidase by subjecting them to the same conditions as used for derivation of the original released proteome, namely incubation in Creek's medium without serum for 2 h, followed by centrifugation to generate the cell pellet and culture supernatant. The resulting material was then analysed for the presence of each tagged peptidase in the pellet or supernatant or for the presence of the abundant cytosolic protein EF1-alpha, which was anticipated to remain cell associated if the cells in the assay remained intact. Figure 2a demonstrates that for every peptidase, the tagged protein was distributed between the supernatant and pellet fractions whereas the EF1-alpha signal was exclusively restricted to the pellet, demonstrating cell integrity. This confirmed the peptidases were indeed released from bloodstream form parasites when expressed at physiological levels under their own 3′UTR gene expression control signals. Moreover, the presence of an N-terminal epitope tag suggested a signal sequence was not contributing to their release.

**Screening released peptidases for enhanced development upon ectopic overexpression.** To assess whether any of the identified proteases contributed to the generation of the parasite QS signal, each of the released peptidases was engineered for doxycycline-inducible ectopic overexpression. Thus, genes encoding an N-terminal epitope-tagged copy of each peptidase were introduced into a regulatable ectopic expression system in *T. brucei* EATRO 1125 AnTat1.1 90:13, parasites that are pleomorphic in vivo and so retain their capacity for QS-driven stumpy formation. Once generated, the cell lines were initially evaluated for the inducible expression of each tagged protein and monitored for their extracellular release. Although cell lines able to inducibly express two of the peptidases (Tb927.10.12260; Tb927.1.2100) could not be selected, ten peptidases were successfully expressed, each exhibiting doxycycline-inducible expression of the tagged protein (Fig. 2b). Quantification of the ectopic expression of Tb927.11.12850 and Tb927.11.2500 relative to their endogenously tagged version of the protein showed 5.6 and 6.3 fold increase respectively (Supplementary Fig. 7b). Furthermore, in each case, the expressed proteins were detected as released from the cells after incubation in serum-free Creek's medium, contrasting with the cytosolic control protein EF1-alpha (Fig. 2b, 'S' fraction).

The generated cell lines were then assayed in biological triplicate for phenotypes resulting from their overexpression in vitro, with a particular focus on any effects on cell growth. Supplementary Fig. 2 shows the growth profile of each peptidase-expressing line along with blots confirming the stringent inducible regulation of expression of the tagged proteins. Of the ten peptidases analysed, ectopic overexpression of two, Tb927.8.8330 and Tb927.11.6590, substantially reduced growth (>50%) upon induction, with the former resulting in rapid cell death of the parasites (Supplementary Fig. 2a). Induction of the expression of Tb927.6.400, Tb927.8.7020 and Tb927.11.12850 reduced the growth of the cells somewhat (10–50% growth reduction) (Supplementary Fig. 2b), whereas Tb927.3.4750

enhanced cell growth when expressed (Supplementary Fig. 2c). The remaining four peptidases had minimal effects (<10%) on the growth of the parasite populations (Supplementary Fig. 2d).

Having observed effects in vitro we monitored the consequences for parasite differentiation to stumpy forms of ectopic overexpression of each peptidase in vivo. Thus, each of the peptidase expressing lines was analysed in replicate mouse infections, with peptidase expression induced, or not, by provision of doxycycline or 5% sucrose, respectively, to the drinking water of infected animals. Given its rapid death in vitro, Tb927.8.8330 was not included in the analysis. Infections were monitored over 6 days, with a reduced parasitaemia progression potentially indicative of developmental acceleration in response to the peptidase ectopic overexpression. Figure 3a demonstrates that three peptidases of 9 analysed resulted in substantially premature growth arrest of the parasites in vivo, these being Tb927.8.7020 (Peptidase 1), Tb927.11.12850 (Oligopeptidase B) and Tb927.11.2500 (Metallocarboxypeptidase 1). Despite the reduced parasitaemia, the induced parasites were morphologically stumpy and the expression of the stumpy specific marker protein PAD1[19] was detectable (supplementary Fig. 7a). This revealed premature differentiation to stumpy forms was elicited by enhanced expression of each of the three peptidases. Tb927.11.6590 also showed reduced growth reflecting its slow growth in vitro (Supplementary Fig. 2a).

For all peptidases, the relative level of expression of each, and their relative extracellular release, were compared (Fig. 3b). We observed no correlation between those peptidases that induced enhanced differentiation when induced (Fig. 3a) and their relative expression or release as assayed by western blotting (Fig. 3b). Hence, the premature differentiation that was observed was not a consequence of their higher expression relative to other peptidases.

**Peptidase expression enhances differentiation via the QS pathway.** We next examined whether the accelerated differentiation caused by expression of the peptidases was stimulated through the characterised QS signalling pathway. To achieve this, the inducible ectopic expression of each peptidase that promoted accelerated development (Tb927.8.7020, Tb927.11.12850, Tb927.11.2500) was engineered in a cell line deleted for the QS signalling pathway components RBP7A and B, which are tandemly arranged[7,8]. Deletion of both copies of this gene renders parasites less able to respond to the QS signal in vivo and thereby generate stumpy forms. Figure 4 shows the infection profile of parasites in mice (Fig. 4a) with ectopic expression of each peptidase induced or not in the RBP7 null mutant lines. In contrast to cells with an intact QS signalling pathway (Fig. 3), in this case, the induction of the peptidases did not strongly reduce the parasitaemia in vivo and the infections progressed with limited detectable generation of morphologically stumpy forms in either induced or uninduced parasite lines. To ensure that the peptidases were effectively produced and released in the RBP7 null mutant lines, their regulated expression was examined by western blotting, demonstrating inducible expression of each peptidase and its presence in both the cell pellet and supernatant, confirming extracellular release (Fig. 4b). Despite the high parasitaemias of each infection (exceeding $1 \times 10^9$ parasites/ml), only a low level of PAD1 expression was observed by western blotting (Fig. 4b, Supplementary Fig. 7c) and immunofluorescence (Fig. 4c) and this was not significantly different regardless of the expression of the peptidases.

**Peptidase null mutants exhibit delayed differentiation.** Having demonstrated that the ectopic overexpression of three peptidases promoted accelerated differentiation of trypanosomes in vivo, we

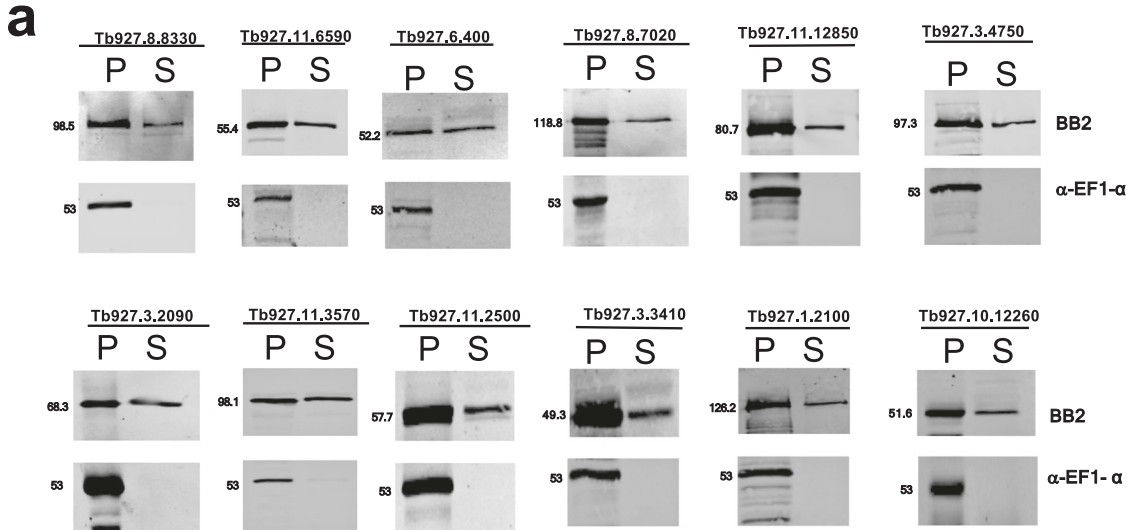

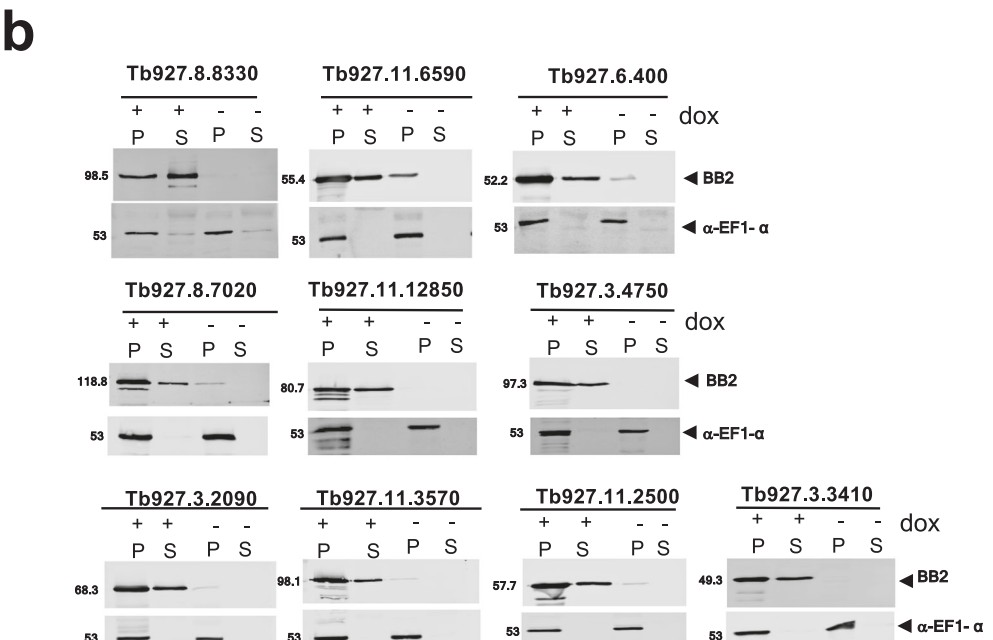

**Fig. 2 Peptidases identified by mass spectrometry are released from intact trypanosomes. a** Release of identified peptidases by bloodstream form parasites. In each case, the identified peptidase was epitope tagged using constructs targeting the endogenous gene locus to ensure close to physiological expression. Resultant cell lines were assayed in serum-free Creek's medium for 2 h and then the cell pellet (P) and supernatant (S) isolated by centrifugation. Isolated proteins were monitored for the distribution of each peptidase using the BB2 antibody detecting the Ty1 epitope tag, with the abundant cytosolic protein EF1-alpha used to monitor non-specific escape of cytosolic proteins. Molecular weight markers in kDa. **b** Inducible ectopic expression of each peptidase identified as released by parasites. In each case, N-terminally epitope-tagged genes were cloned into a doxycycline-inducible expression vector and then the expression and extracellular release of the expressed proteins were monitored using the epitope tag-specific antibody BB2, with the cytosolic protein EF1-alpha used as a control. Released and cell-associated proteins were isolated after incubation of parasites in serum-free Creek's medium for 2 h and centrifugation. Molecular weight markers in kDa. Source data are provided as a source data file.

explored whether the individual deletion of the genes for each peptidase would reduce differentiation efficiency. Therefore, we exploited CRISPR-mediated gene replacement in the *T. brucei* EATRO 1125 AnTat 1.1 J1339 cells line which constitutively expresses CAS9[9], to create null mutants for the identified peptidases. Although null mutants for Tb927.8330, Tb927.3.2090 and Tb927.10.12260 were not successfully isolated, null mutants for nine peptidases were generated. This included the peptidases Tb927.8.7020 (peptidase 1), Tb927.11.2500 (metallocarboxypeptidase 1) and Tb927.11.12850 (oligopeptidase B), which generated accelerated differentiation when overexpressed. Once gene deletion was confirmed for each of these peptidases (Fig. 5a shows the QS-active peptidases), the respective knock-out lines were compared to wild-type parents in order to assess the parasite growth and level of differentiation to stumpy forms. For all nine peptidase null mutants, growth in vitro was similar to

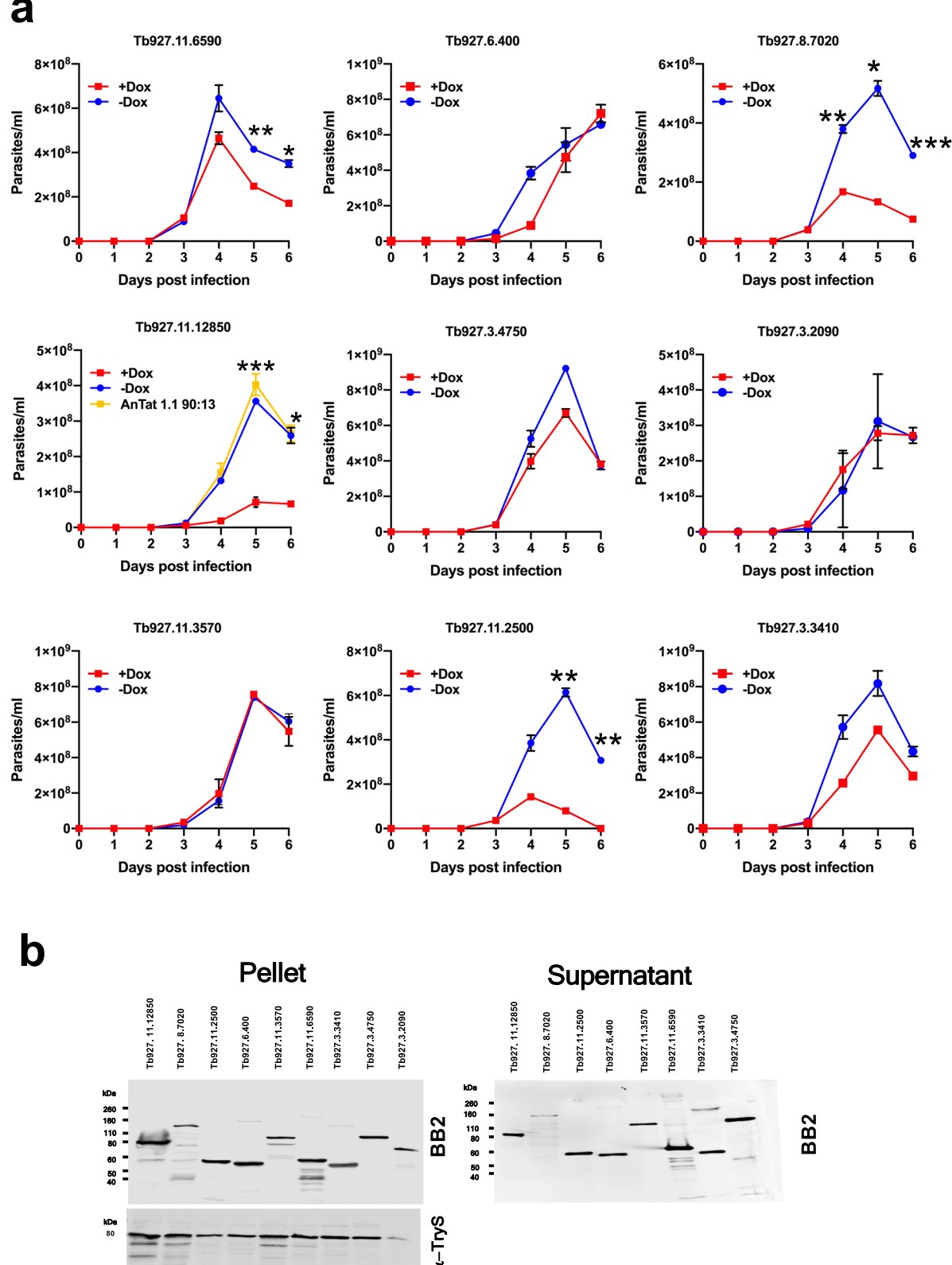

wild type cells (Supplementary Fig. 3) and for six peptidases, including Tb927.8.7020, the progression of the parasitaemia in mice was equivalent to parental cells, although Tb927.11.6590 slightly elevated the overall parasitaemia (Fig. 5b and Supplementary Fig. 4). This indicated their individual deletion did not strongly affect the fitness or the developmental progression of the parasites in vivo. In contrast, null mutants for both Tb927.11.2500 and Tb927.11.12850 both showed a higher total parasitaemia on day 4 post infection compared to parental cells indicating reduced differentiation to stumpy forms in vivo (Fig. 5b). Nonetheless, although delayed, in both cases the cell lines developed stumpy forms by day 5 of infection, supportive of the signal being generated through the action of more than one peptidase.

**Fig. 3 Ectopic expression of three peptidases enhances QS. a** Ectopic overexpression phenotypes of identified peptidases in vivo. $1 \times 10^4$ of each parasite line were inoculated into MF1 mice and progression of the parasitaemia monitored after induction of the ectopic peptidase expression from the day of infection. For each peptidase, mice were infected, being either induced or uninduced to express the peptidase via provision of doxycycline in 5% sucrose (induced; red) or 5% sucrose (uninduced; blue) to the drinking water of infected animals ($n = 2$ biologically independent analyses, with $n = 3$ where effects were seen in trial studies). The parasitaemia of the parental *T. brucei* AnTat 1.1 90:13 cells was also assayed (yellow) and is shown in panel Tb927.11.12850 to illustrate its equivalent progression to uninduced parasites. Values are mean ± SEM. Two-way Anova followed by Tukey's or Šídák's multiple comparison test, $p < 0.05$ (*); $p < 0.002$ (**); $p < 0.0002$ (***) $p < 0.0001$ (****). Tb927.11.6590; $p = 0.0092$ and 0.0295 respectively, Tb927.8.7020; $p = 0.0070$, 0.0207 and 0.0004 respectively, Tb927.11.12850; $p = 0.0004$ and 0.0177 respectively, Tb927.11.2500; $p = 0.0085$ and 0.0038 respectively. **b** Relative expression and extracellular release of each identified peptidase. In each case, cell pellet and supernatant were isolated after 2 h incubation in Creek's medium and the epitope-tagged peptidase was detected using the Ty1-specific BB2 antibody. Loading is indicated by the cytosolic protein Trypanothione synthetase (TyrS). The expression level and extracellular release of the respective peptidases did not correlate with phenotypic effects observed either in vitro or in vivo. Source data are provided as a source data file.

To confirm that the reduced differentiation was a consequence of the gene deletion of the peptidases Tb927.1.12850 and Tb927.11.2500, we transfected the null mutant lines to restore expression of each peptidase at its endogenous locus. Once the reintegration of each peptidase gene was validated (Fig. 5c shows Tb927.10.12850; Supplementary Fig. 5 shows Tb927.11.2500) these 'add back' cell lines were assayed for their growth and differentiation in vivo in comparison to the parental cell line and each null mutant. Figure 5c and d shows that add back of the peptidase Tb927.11.12850 restored differentiation to levels observed in the parental cell lines, with PAD1 expression being significantly different between the parental and null mutant at peak parasitaemia (day 5; $p = 0.014$) and between the null mutant and the add back cell line (day 5; $p = 0.007$), but not between the parental and add back lines (Day 5; $p = 0.929$). For Tb927.11.2500, parasites expressing the add back were successfully recovered and the expression of the restored gene was confirmed (Supplementary Fig. 5a, b) and these cells grew in vitro equivalently to wild-type cells (Supplementary Fig. 5c). However, these parasites were unable to establish infections in mice despite the analysis of several cell lines and using alternative routes of infection (intraperitoneal, intravenous). Consequently, the restoration of stumpy formation could not be unequivocally confirmed in this case.

**Knockout of two peptidases combinatorially reduce QS signalling.** Having established that deletion of two peptidases, Tb927.11.12850 (oligopeptidase B) and Tb927.11.2500 (metallocarboxypeptidase 1) reduced differentiation individually we sought to establish if they would operate combinatorially to contribute to the generation of the QS signal. To explore this, a parasite line was created that had deleted both alleles of both peptidase genes (Tb927.11.12850::Tb927.11.2500) and this line was then compared with the parental line and one of the individual gene knockout lines (Tb927.11.12850 KO) for their growth and differentiation in vivo. Figure 6a demonstrated that the deletion of both peptidases generated a parasitaemia of enhanced virulence, exceeding even that seen where one peptidase (Tb927.11.12850) alone was absent (Supplementary Table 1). Supporting this being a consequence of reduced differentiation efficiency, analysis of the expression of PAD1 revealed that both the single and double peptidase deletion expressed less PAD1 at the peak of the infection on day 5 than the parental line although the single and double peptidase deletion lines were not significantly different from each other with respect to this marker ($p = 0.933$) (Fig. 6b). To confirm that the observed effects on differentiation were not a consequence of multiple transfection rounds of the parasites, Tb927.11.12850 was reintroduced into the double knockout line and their parasitaemia and differentiation examined. Figure 6c demonstrated that re-introduction of the Tb927.11.12850 reduced parasite virulence compared to the

double KO mutant (Supplementary Table 1), restoring it to levels similar to the individual knockout of Tb927.11.12850. As expected, PAD1 levels remained low, as with the single knockout lines. On the basis of these data, we conclude that the combined activity of both peptidases, oligopeptidase B and metallocarboxypeptidase 1, contributes to the parasite's capacity for QS, rather than there being complete redundancy between their action.

**Peptidase release is mediated via unconventional protein secretion.** The protein sequences of the released peptidases characterised in our analyses did not predict the presence of a secretory signal sequence on the N- terminus of the respective proteins. Therefore, we wished to explore whether the QS active peptidases were being released by Golgi-ER mediated classical secretion or by unconventional protein secretion, such as via extracellular vesicles or other uncharacterised pathways. To explore this, the cytological distribution of the peptidases that promoted differentiation was initially assessed via immunofluorescence assay and cell fractionation (Supplementary Fig. 6). This revealed that both exhibited a cytoplasmic location distinct from the ER as visualised by the ER marker BiP, and this was consistent with their enriched fractionation with cytosolic proteins compared with organellar or nuclear protein markers, and with earlier studies in *T. b. evansi*[20]. To explore the mechanism of release in more detail for oligopeptidase B, cell lines were generated that expressed C-terminally epitope tagged Tb927.11.12850 in a cell background where a dominant-negative mutant of TbSAR1 (TbSAR1-DN), or the wild type protein (TbSAR1-WT) was inducibly expressed, or where Rab11 was inducibly silenced by RNAi. Expression of TbSAR1-DN inactivates the classical secretory pathway by preventing trafficking from the Golgi to the ER whereas Rab11 depletion increases nanotube and EV mediated protein release[21]. In contrast to expression of TbSAR1-WT, when TbSAR1-DN was inducibly expressed, cell growth stopped within 12–24 h confirming effective expression of the dominant-negative protein (Fig. 7a), this being consistent with the timescale of previous studies[22]. Analysis of the extracellular release of Tb927.11.12850 during the first 18 h of TbSAR1-DN expression however did not reduce the peptidase detected in the culture supernatant compared to the TbSAR1-WT expressing line, despite cell integrity being sustained on the basis of EF1-alpha cell-association (Fig. 7b). Induction of RNAi against TbRab11 also generated a rapid cessation of cell growth (within 24 h) (Fig. 7c) but here Tb927.11.12850 also continued to be released at levels equivalent to the uninduced cells (Fig. 7d).

In combination, this indicated that trypanosome Oligopeptidase B release was not directed through either the classical secretory pathway or through extracellular vesicle shedding. Instead, an alternative unconventional protein secretion mechanism is invoked.

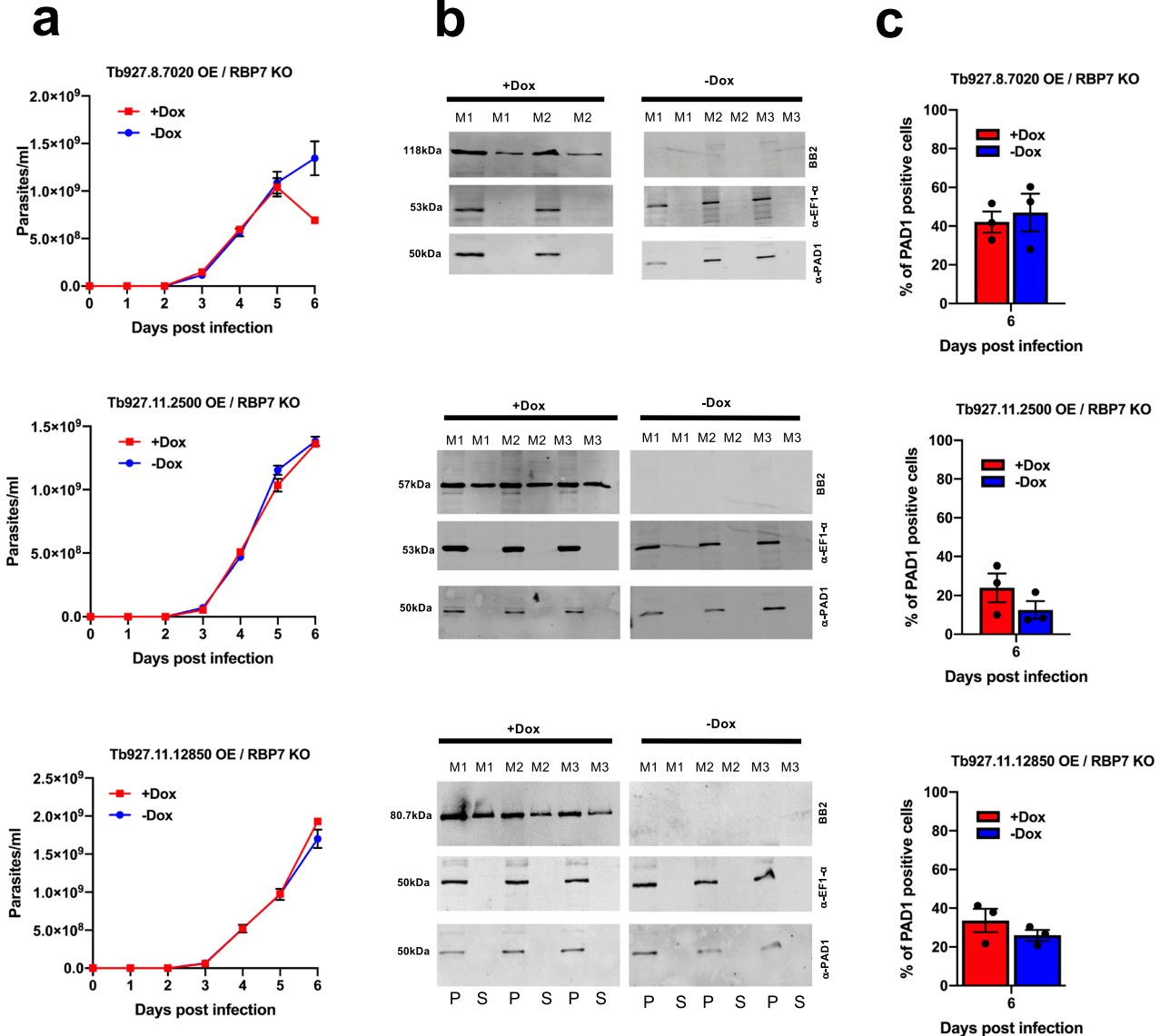

**Fig. 4 Accelerated differentiation activated by the expression of three peptidases is mediated via the QS signalling pathway. a** Infection profile for mice infected with parasite lines ectopically expressing each peptidase (Tb927.11.2500, Tb927.11.12850 or Tb927.8.7020) under doxycycline regulation with the QS signalling pathway disrupted (ΔRBP7A/RBP7B). In each case, six mice were infected (biologically independent samples), with three being provided with doxycycline to induce ectopic peptidase expression (uninduced, blue; induced, red). In all cases, the parasitaemia (mean ± SEM) progressed unchecked due to the lack of transduction of the QS signal caused by deletion of RBP7A/B. **b** Confirmation of the inducible expression and extracellular release of each peptidase in an ΔRBPA/B background. Parasites were harvested from the infections shown in **a**, with three mice being induced for ectopic peptidase expression and three being uninduced as biologically independent samples. Detection of the stumpy marker PAD1 demonstrated equivalent stumpy generation in the parasites regardless of the inducible expression of the peptidase. Stringent doxycycline-regulated expression is observed in each case. The loading control for the cell pellet was EF1-alpha. Parasites were not successfully purified from one of the mice in the +Dox group at the end of the experiment due to a failure with the DEAE resin. **c** Quantitation of PAD1 expression on day 6 post infection when parastaemias exceeded $1 \times 10^9$/ml. PAD1 immunofluorescence assays were based on three biologically independent infection profiles represented in **a** and analysed by western blotting represented in **b**. Source data are provided as a source data file.

## Discussion

We have explored released peptidases that could contribute to the generation of the trypanosome's oligopeptide QS signal using a combination of systematic ectopic overexpression and gene deletion. Our analyses identified 12 released peptidases most of which could be deleted without consequence for the parasites in vitro or in vivo. This is consistent with the analysis of all trypanosome peptidases by ref. [15], where all but one peptidase was found to be non-essential in a systematic screen, as assessed by RNAi. However, that analysis used monomorphic parasites that have lost the ability to become stumpy and the study was

mostly restricted to phenotypic analysis in vitro. By using pleomorphic cell lines able to generate stumpy forms and a combination of ectopic overexpression and gene deletion using CRISPR Cas9 we discovered via in vivo analysis that two released peptidases, oligopeptidase B (Tb927.11.12850) and metallocarboxypeptidase 1 (Tb927.11.2500), significantly and individually modulate the levels of parasite QS in vivo, suggesting specificity in their QS signal generation. We have previously also shown that prolyl oligopeptidase and pyroglutamyl peptidase can promote parasite differentiation when ectopically overexpressed, but these were not detected as components of the secreted material released

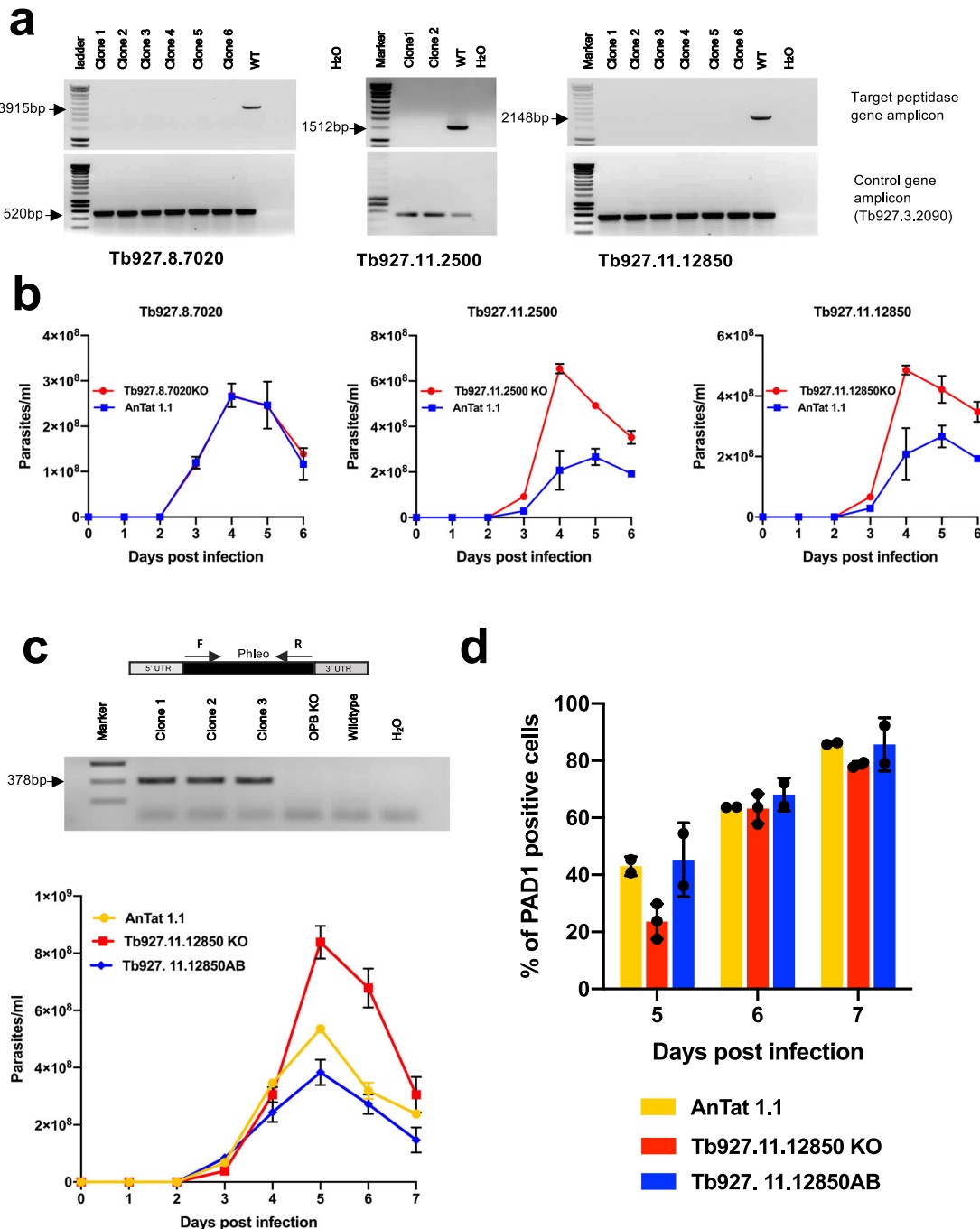

**Fig. 5 Null mutants for two released peptidases show reduced quorum sensing. a** PCR analysis of genomic DNA derived from null mutants of Tb927.8.7020, Tb927.11.2500 and Tb927.11.12850 analysed using gene-specific primers. Several null mutant clones were analysed in each case, together with wild-type parental parasites. For each target gene no amplicon was detected with the null mutant clones, whereas a band of the expected size was detected using gDNA from wild-type parasites. Integrity of the gDNA in each case was validated using primers targeting a control gene (lower panels). **b** Parasitaemias (mean ± SEM) of infections initiated with null mutant line (red) or parental line (blue) for each target gene. Biologically independent replicate infections are shown ($n = 2$), with Tb927.11.2500 and Tb927.11.12850 showing elevated parasitaemia. Note that the scales differ on the y axis between graphs. **c** Upper panel- PCR confirmation of the successful add back of a copy of Tb927.11.12850 to the null mutant. Lower panel- parasitaemias (mean ± SEM) of biologically independent infections initiated with parental *T. brucei* AnTat 1.1 parasites (yellow, $n = 2$), a Tb927.11.12850 null mutant (red, $n = 3$), or the Tb927.11.12850 null mutant containing an add back for the gene (blue, $n = 2$). The null mutant exhibited elevated parasitaemia reflecting reduced differentiation, whereas the add back of the Tb927.11.12850 gene restored the growth profile to levels similar to the parental line. **d** PAD1 expression of parasites at day 5, 6 and 7 post infections. The null mutant exhibited reduced PAD1 expression at peak parasitaemia compared to the parental cells or add back line in biologically independent analyses (respectively $n = 3$, $n = 2$, $n = 2$). Beyond day 5, PAD1 levels were similar between lines, reflecting that the null mutant exhibited delayed but not abrogated stumpy formation, with stumpy parasites further enriched due to their enhanced tolerance of the developing immune response beyond day 5. Source data are provided as a source data file.

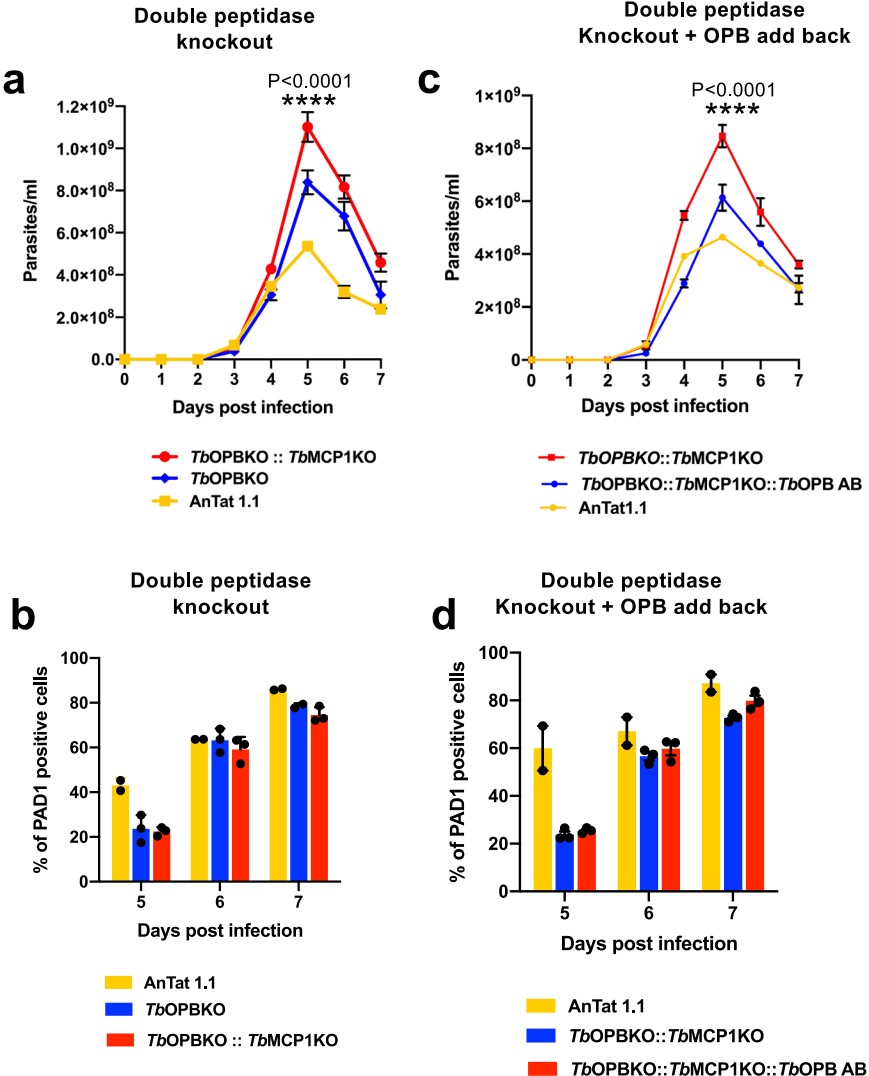

**Fig. 6 Two released peptidases contribute combinatorially to the generation of the quorum sensing response. a** biologically independent infection profiles (mean ± SEM) of parental *T. brucei* AnTat 1.1 parasites (yellow, $n = 2$), Tb927.11.12850 null mutants (blue, $n = 3$) and Tb927.11.12850 plus Tb927.11.2500 double null mutants (red, $n = 3$) in vivo. The Two-way ANOVA analysis followed by Tukey's comparison test was performed for statistical significance between Tb927.11.12850KO vs Tb927.11.12850 plus Tb927.11.2500 double null mutant (Supplementary Table 1). The double peptidase KO was most virulent, the single peptidase KO was less virulent but more virulent than the parental line. Two-way ANOVA analysis followed by Tukey's comparison test was performed, $P < 0.0001$ (****). **b** PAD1 expression (mean ± SEM) of *T. brucei* AnTat 1.1 parasites, Tb927.11.12850 null mutants and Tb927.11.12850 plus Tb927.11.2500 double null mutants on day 5, 6 and 7 of infection (respectively, $n = 2$, $n = 3$, $n = 3$ in biologically independent samples). At peak parasitaemia on day 5 both the single peptidase and double peptidase infections exhibited less PAD1 expression than parental cells. Beyond day 5, PAD1 levels were similar between lines, reflecting that the null mutants exhibited delayed but not fully abrogated stumpy formation, with stumpy parasites further enriched due to their enhanced tolerance of the developing immune response beyond day 5. **c** biologically independent infection profiles (mean ± SEM) of parental *T. brucei* AnTat 1.1 parasites (yellow, $n = 2$), Tb927.11.12850 plus Tb927.11.2500 double null mutants (red, $n = 3$), and Tb927.11.12850 plus Tb927.11.2500 double null mutants with a restored Tb927.11.12850 gene (blue, $n = 3$) in vivo. The statistical significance between Tb927.11.12850 plus Tb927.11.2500 double null mutant and the restored Tb927.11.12850 cell line is shown ($p < 0.0001$). The double peptidase KO was most virulent, whereas the add back less virulent than the double KO but more virulent than the parental cell line. Two-way ANOVA analysis followed by Tukey's comparison test was performed, $P < 0.0001$ (****) (Supplementary Table 1). **d** PAD1 expression (mean ± SEM) of *T. brucei* AnTat 1.1 parasites, Tb927.11.2500::Tb927.11.2500 double null mutants and Tb927.11.2500::Tb927.11.2500 double null mutants with restored Tb927.11.12850 add back on day 5, 6 and 7 of infection (respectively, $n = 2$, $n = 3$, $n = 3$ in biologically independent samples). At peak parasitaemia on day 5 both the double peptidase null mutants and the add back mutant infections exhibited less PAD1 expression than parental cells. Beyond day 5, PAD1 levels were similar between parasite lines, reflecting that the null mutants and add back line exhibited delayed but not fully abrogated stumpy formation, with stumpy parasites further enriched due to their enhanced tolerance of the developing immune response beyond day 5. Source data are provided as a source data file.

by bloodstream from parasites in the current study using the applied thresholds. Similarly, we show here that Peptidase 1 (Tb927.8.7020) can enhance differentiation when ectopically overexpressed although its deletion does not detectably influence stumpy formation. Hence, several peptidases could contribute to

production of the QS signal with considerable redundancy. However, only oligopeptidase B and metallocarboxypeptidase 1 null mutations affected overall development of the parasites in the bloodstream, despite the potential for oligopeptidase B null mutants to exhibit a compensatory elevated prolyl oligopeptidase

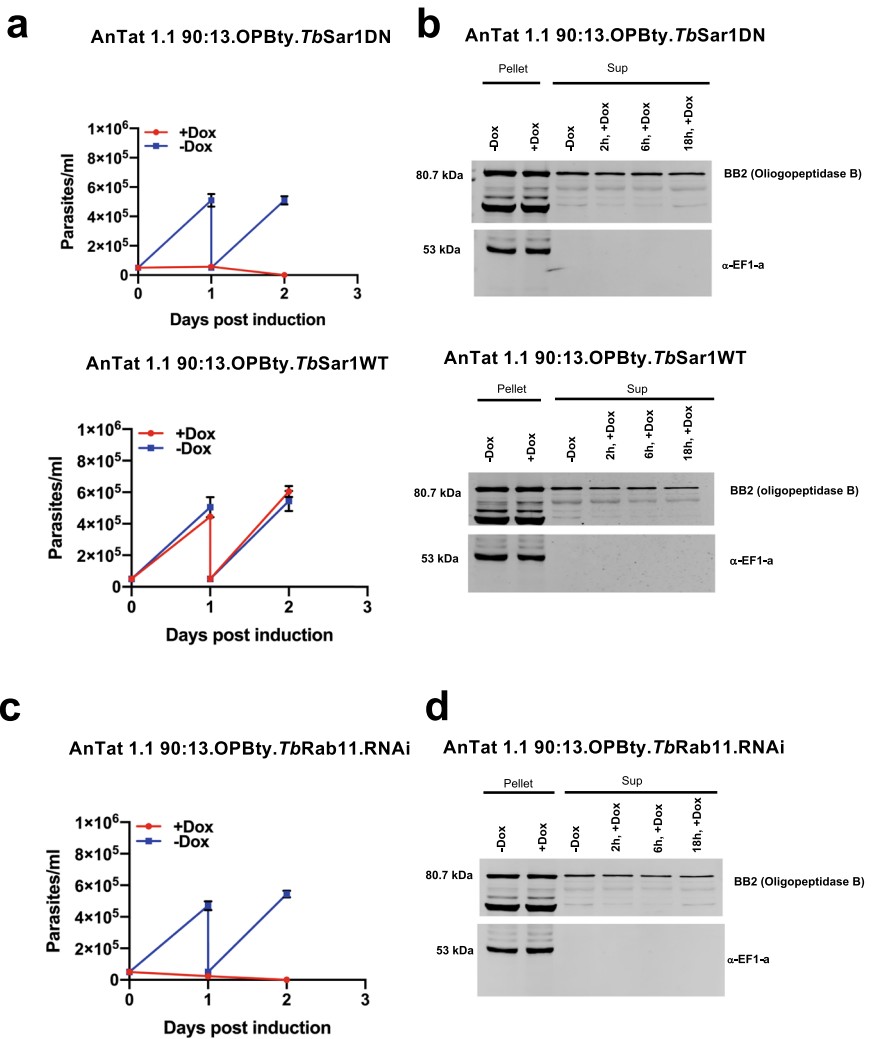

**Fig. 7 Oligopeptidase B exhibits unconventional protein secretion. a** Growth in vitro (mean ± SEM) of *T. brucei* AnTat1.1 90:13 constitutively expressing epitope tagged Tb927.11.12850 (oligopeptidase B), induced to express dominant-negative SAR1 (upper panel) or wild type SAR1 (lower panel)($n = 2$ in biologically independent analyses). Expression of the dominant-negative SAR1 induced growth cessation within 24 h. **b** Extracellular release of epitope tagged Tb927.11.12850 from parasites induced to express dominant-negative SAR1 (upper panel) or wild type SAR1 (lower panel). Tb927.11.12850 was released at equivalent levels in each case (with an additional degradation product in the pellet fractions only), whereas the cytosolic control EF1-alpha remained cell associated. **c** Growth in vitro (mean ± SEM) of *T. brucei* AnTat 1.1 90-:13 constitutively expressing epitope tagged Tb927.11.12850, with induced RNAi targeting Rab11 ($n = 2$ in biologically independent analyses). RNAi against Rab11 induced growth cessation within 24 h. **d** Extracellular release of epitope tagged Tb927.11.12850 from parasites induced to deplete Rab11 by RNAi. Extracellular release of Tb927.11.12850 was sustained at equivalent levels, whereas the cytosolic control EF1-alpha remained cell associated. Source data are provided as a source data file.

activity[23]. Thus, the analysis of individual and combinatorial phenotypes resulting from gene knockout of oligopeptidase B and metallocarboxypeptidase 1 identifies that these molecules are the dominant contributors to the generation of the trypanosome QS signal although other peptidases could provide additional minor contributions (Fig. 8).

Several analyses of the molecules secreted by African trypanosomes have identified that peptidases are commonly detected in the excretory/secretory material released by parasites in their mammalian host. A comprehensive literature review of these activities released by *Trypanosoma brucei gambiense* provided in ref. 16 describes 444 proteins in 12 functional classes, with 10 peptidase families or subfamilies represented. Of the released molecules 134 were bloodstream specific and 10% of those were peptidases. It has been proposed that these molecules contribute to the degradation of peptide hormones contributing to the physiological effects of trypanosome infection. Of those analysed, oligopeptidase B[24] and prolyl oligopeptidase[11] have been shown

to retain activity in the bloodstream, with oligopeptidase B able to cleave atrial natriuretic factor, vasopressin and neurotensin[20]. In contrast, metallocarboxypeptidase was proposed to modulate the level of vasoactive kinins[25]. The activity of metallocarboxypeptidase 1 was also proposed to act on proteins of the extracellular matrix and it has been suggested their activity against the blood–brain barrier could contribute to the parasite's tropism.

Previous analyses of the release of proteins by African trypanosomes recognised the absence of a signal sequence on many of the molecules identified[26]. Classical secretion involves recognition of an N terminal leader sequence and then trafficking from the Golgi through the ER to the flagellar pocket for ultimate release[27]. This process can be disrupted by the expression of dominant-negative SAR1 that blocks trafficking between the Golgi and ER, and is rapidly lethal[22]. Alternatively, proteins may be released from parasites via nanotube shedding in a process that is enhanced with Rab11 depletion but not signal sequence dependent[21]. The release of the Oligopeptidase B identified in this

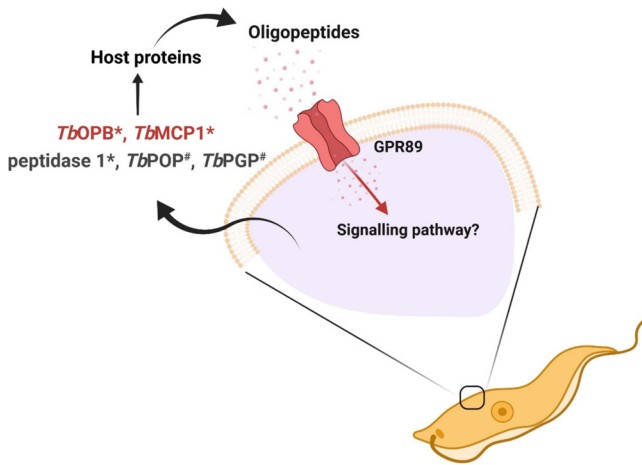

**Fig. 8 A model for the quorum-sensing signalling in trypanosomes.**
During trypanosome infection, the parasites release peptidases into the
bloodstream of the mammalian hosts. These hydrolyse host proteins,
generating oligopeptides which are transported by GPR89, a surface
transporter protein found only on the surface of the slender forms of
trypanosome parasites. * peptidases identified in this study, # peptidases
shown to enhance stumpy differentiation by ectopic overexpression in
ref. [9]. TbOPB is trypanosome oligopeptidase B (Tb927.11.12850), TbMCP1 is
metallocarboxypeptidase 1 (Tb927.11.2500), Peptidase 1 is Tb927.8.7020,
TbPOP is prolyl oligopeptidase (Tb927.10.8020) and TbPGP is
pyroglutamyl peptidase (Tb927.4.2670).

study did not show evidence of release by either route, suggesting
an unconventional protein secretion (UPS) pathway contributes.
UPS has been identified as a significant component of extra-
cellular protein release for molecules involved in parasite-host
interactions, examples including ABC transporter-mediated
release of acylated parasite proteins (e.g. Leishmania HASP
secretion;[28]) autophagosome release or other uncharacterised
UPS routes[17]. Supporting a non-ER-based release of the QS-
peptidases identified here, both Oligopeptidase B and Metallo-
carboxypeptidase 1 exhibited cytosolic distribution by immuno-
fluorescence microscopy and cell fractionation.

Trypanosomes undergo stumpy formation in the bloodstream
but also adipose tissue and skin, and in each compartment, they
can comprise the dominant developmental form in chronic
infections[6,29,30]. In the bloodstream, stumpy formation is related
to parasite density due to the enhanced QS signal generation
provided by large numbers of parasites releasing peptidases and
so generating the oligopeptide signal driving the developmental
response[31]. In livestock infections, however, parasitaemias are
lower and parasite numbers may not drive the QS response in the
bloodstream as effectively as in high parasitaemia experimental
infections in mice[32]. However, for parasites constrained in the
tissues, absolute parasite numbers are less relevant for QS than
the local concentration of parasites and their released peptidases,
which can generate oligopeptide signals proximally to the com-
partmentalised parasites[9,12]. In this low flow environment, QS
can generate stumpy forms at low parasitaemia and as a high
proportion of the local parasite concentration. In this scenario,
stumpy forms dominate irrespective of the blood parasitaemia
favouring parasite transmission, resolving the so-called 'trans-
mission paradox' where parasite spread is maintained despite low
overall parasite numbers in the host as determined by blood-
stream parasitaemias[33]. These local stumpy populations, observed
in skin and adipose tissue, have the ability to sustain disease
spread because small numbers of parasites can infect tsetse flies,
at least under laboratory conditions using permissive flies[34,35].

Our discovery that oligopeptide signals from the action of oli-
gopeptidase B and metallocarboxypeptidase 1 in infections pro-
vide a coherent picture of how parasites constrained in the tissues
can promote the formation of transmissible stumpy forms,
including within a tissue environment able to support disease
spread within the physiological context of parasites in their nat-
ural host setting.

## Methods

**Trypanosome cell lines**. *T. brucei* EATRO (East African Trypanosomiasis
Research Organisation in Tororo, Uganda) 1125 AnTat 1.1 90:13 cells and AnTat
1.1 J1339, both pleomorphic cell lines, were used in this study. Pleomorphic cell
lines are able to differentiate to the various life cycle forms of the parasite.

The pleomorphic bloodstream forms of the parasite were cultured in HMI-9
medium supplemented with 20% v/v Foetal Bovine Serum (FBS) at 37 °C with 5%
$CO_2$. They were maintained between $1 \times 10^5$ and $1 \times 10^6$ cells/ml and passaged
every 24 to 48 h in a vented culture flask to allow the diffusion of $CO_2$ into
the flask.

**Mouse infections**. Female MF1 mice older than 6 weeks were used for all
experiments and sourced either from Charles River or locally bred. They were kept
at an ambient temperature of 21 °C and 56% humidity with 12 h of light and 12 h
of dark cycle. Mice were inoculated with various *T. brucei* cell lines intraper-
itoneally (10,000 parasites) and the parasitaemia was monitored daily from day
three post-infection. The appropriate life cycle forms of the parasite were harvested
by collecting blood from trypanosome-infected mice by cardiac puncture. The
parasites were then purified by separation on a diethyl aminoethyl cellulose anion
exchange column. The parasites were subsequently counted using the Neubauer
haemocytometer and then washed with Phosphate Buffered Saline-Glucose (PSG)
(44 mM NaCl; 57 mM $Na_2HPO_4$ 3 mM $KH_2PO_4$; 55 mM glucose, pH 7.8) by cen-
trifugation at $1600 \times g$ for 10 min. The cells were resuspended in an appropriate
volume of pre-warmed HMI-9 media.

All work was carried out under a UK home office licence (P262AE604) that had
been approved after local ethical review at the University of Edinburgh Animal
Welfare Ethical Review Body and UK Home Office.

**Detection of released extracellular proteins**. To isolate released parasite-derived
material from cultured bloodstream forms, $2 \times 10^7$ trypanosomes were washed with
PSG at $1600 \times g$ for 5 min twice and incubated in Creek's Minimal Medium
(CMM) without serum for 2 h at 37 °C, 5% $CO_2$. After the 2-h incubation, the cells
were centrifuged at $1054 \times g$ for 10 min and the supernatant removed. This was
spun again at $1054 \times g$ for 10 min to ensure the removal of all parasites from the
supernatant. An aliquot of 100 μl of the supernatant was analysed for trypanosome
EF1-alpha (EF1α) using a western blot. The remaining protein was concentrated
using cold acetone precipitation method. Briefly, the supernatant was mixed with
pre-cold acetone 4× the volume of the supernatant and incubated at −20 °C
overnight. This was spun the following day at $15,000 \times g$ for 10 min and the pellet
dried at room temperature for 30 min to allow the acetone to evaporate. The
precipitated protein samples were then submitted to the Edinomics research
facility, the University of Edinburgh for MS/MS analysis. Triplicate samples were
prepared.

To isolate stumpy derived material, the stumpy forms of *T. brucei* EATRO 1125
AnTat 1.1 90:13 were harvested from mouse infections and purified by DE52 anion
exchange chromatography. The parasites were immediately washed with PSG and
incubated in HMI-9 medium supplemented with 20% v/v FBS for 2 h at 37 °C, 5%
$CO_2$ to allow the parasites to adapt to the in vitro condition. After the adaptation
period, $2 \times 10^7$ cells were washed by centrifugation with PSG and incubated in
CMM without serum for 2 h at 37 °C, 5% $CO_2$ to allow for secretion/release of
proteins.

For material derived from parasites undergoing differentiation to procyclic
forms, *T. brucei* EATRO 1125 AnTat 1.1 90:13 stumpy forms harvested and
purified from infected mice were washed with PSG and incubated in HMI-9
medium supplemented with 20% v/v FBS and 6 mM of cis-aconitate to induce
differentiation to the procyclic forms for one and 4 h at 37 °C, 5% $CO_2$. After each
incubation period, $2 \times 10^7$ cells were washed by centrifugation with PSG and
incubated in CMM plus 6 mM cis-aconitate without serum for 2 h at 37 °C, 5%
$CO_2$. The supernatant was collected after the incubation period and processed the
same way as for slender and stumpy forms.

**Mass spectrometry**. Samples were run on a pre-cast Bolt™ 4-12% Bis-Tris Plus gel
(Invitrogen) for 5 min before overnight in-gel trypsin digestion. Peptide extracts
were dried by Speedvac and the dried peptide samples were re-suspended in MS-
loading buffer (0.05% trifluoroacetic acid in water) then filtered using a Millex filter
before HPLC-MS analysis. The analysis was performed using an online system of a
nano-HPLC (Dionex Ultimate 3000 RSLC, Thermo-Fisher) coupled to a QExactive
mass spectrometer (Thermo-Fisher) with a 300 μm × 5 mm pre-column (Acclaim
Pepmap, 5 μm particle size) joined with a 75 μm × 50 cm column (Acclaim

Pepmap, 3 μm particle size). Peptides were separated using a multi-step gradient of 2–98% buffer B (80% acetonitrile and 0.1% formic acid) at a flow rate of 300 nl/min over 90 min. Raw data from MS/MS spectra were searched against a *T. brucei* database using MASCOT (version 2.4). The parameters used in each search were: (i) missed cut = 2, (ii) fixed cysteine carbamidomethylation modification, (iii) variable methionine oxidation modification, (iv) peptide mass tolerance of 10 ppm, (v) fragment mass tolerance of 0.05 Da. Search results were exported using a significance threshold (*p* value) of less than 0.05 and a peptide score cut off of 20.

**Immunofluorescence**. Cells were either fixed as blood/culture smears, air dried on microscope slides and then immersed with ice-cold methanol, or were paraformaldehyde fixed in suspension. Specifically, $2 \times 10^6$ cells were pelleted at $2450 \times g$ for 5 min and washed once with cold PBS. The cells were then fixed by resuspending them in in 125 μl of cold PBS and 125 μl of 8% paraformaldehyde for 10 min on ice. Thereafter, fixed cells were resuspended in 130 μl of 0.1 M glycine in PBS and incubated at 4 °C at least overnight. After this incubation period, the cells were spun, and the pellet resuspended at $1 \times 10^5$ cells per 10 μl of 1× PBS. Cells adhered to Polysine® slides (VWR, 631-0107) in areas demarcated using an ImmEdge hydrophobic barrier pen (Vector Laboratories, H-4000) for 1 h at room temperature in a humidity chamber. After the 1 h incubation, the excess PBS was removed from the wells using an aspirator. For the identification of proteins other than surface proteins, cells were permeabilised with 20 μl of 0.1% triton in PBS for 2 min. The wells were washed with PBS and blocked with 2% BSA for 1 h at room temperature. The slides were then washed in 1× PBS for 2 min and incubated with 50 μl of appropriate primary antibody (diluted in 2% BSA/PBS, 1:5 for BB2 antibody, 1:1000 for PAD1 antibody) for 1 h at 37 °C or 4 °C overnight in a humidity chamber. The slides were washed five times, 5 min each with 1× PBS followed by 50 μl of appropriate secondary antibody incubation (diluted in 2% BSA/PBS, α-mouse Alexa fluor 568 1:500, α-rabbit Alexa fluor 488 1:500). After the secondary antibody incubation, the slides were washed five times, 5 min each and stained with 10 μg/ml 4′,6-diamidino2-phenylindole (DAPI, Life Technologies) for 2 min, then washed with 1× PBS for 5 min and then mounted with 50 μl Mowiol containing 2.5% 1,4-diazabicyclo [2.2.2] octane (DABCO). The slides were covered with coverslips after the Mowiol addition and analysed on a Zeiss Axioskop 2 plus or Zeiss Axio Imager Z2. QCapture Suite Plus Software (version 3.1.3.10, https://www.qimaging.com) was used for image capture and ImageJ for image analysis.

**Western blotting**. Cells were pelleted by centrifugation at $1646 \times g$ for 5 min and then washed with PSG. The resulting cell pellets were resuspended in l00 μl of 1 mM ice-cold Tosyl-L-lysyl-chloromethane hydrochloride and then incubated on ice for 5 min and then at 37 °C for 15 min. After which appropriate volume of Laemmli buffer (62.5 mM Tris-HCl pH6.8, 2% SDS, 10% glycerol, 0.002% Bromophenol blue, 5% β-mercaptoethanol) was added. The samples were either boiled at 95 °C for 5 min to denature the proteins before storage at −20 °C or were directly stored at −20 °C until used.

Proteins resolved by SDS PAGE were transferred to a nitrocellulose transfer membrane (GE Healthcare Amersham™ Protran™) at 80 V for 75 min or 15 V overnight at 4 °C using a Bio-Rad wet transfer apparatus. The membranes were then stained with ponceau for about five mins to ensure the proteins were properly transferred and then washed with distilled $H_2O$ several times. The membranes were blocked with LI-COR Odyssey blocking buffer for one hour at room temperature or 4 °C overnight and then probed with the appropriate primary antibody for 1 h 30 min at room temperature or 4 °C overnight (anti-PAD1; 1/1000; anti-EF1alpha, 1:7000; anti-Ty1/BB2, 1:5). The membranes were washed three times, 10 min each in Tris-buffered saline with 0.05% Tween (TBST) and then incubated in an appropriate secondary antibody (Anti-rabbit (goat anti-rabbit IgG (H + L) Dylight 800; Thermofisher Cat#SA5-10036; used at 1:5000) for 1 h at room temperature, diluted in 50% LI-COR Odyssey blocking buffer and 50% TBST. The membranes were further washed three times, 10 min each in TBST and the proteins visualised by scanning the blots using a LI-COR Odyssey Imager, which uses an infra-red laser to detect the fluorochrome on the secondary antibody.

**Gene manipulation using CRISPR-Cas9**. Gene-specific primers for the amplification of the pPOT plasmid and the small guide RNA (sgRNA) scaffold (G00) were designed using the programme at (www.leishgedit.net/Home.html). 0.5 μM of gene-specific reverse and forward primers, 0.2 mM of dNTPs, PHIRE II polymerase were mixed in 1× PHIRE II reaction buffer, the total volume of 150 μl. PCR cycling conditions were 98 °C for 30 s followed by 98 °C for 10 s, 59 °C for 30 s, 72 °C for 15 s for 35 cycles and then a final extension at 72 °C for 5 min. The precipitated material was then used for parasite transfection.

$3 \times 10^7$ cells were used for each transfection. For CRISPR/Cas9 technology, cell lines constitutively expressing cas9 enzymes were used. Cells were centrifuged at $1600 \times g$ for 5 min and washed by centrifugation with 1 ml of 1× Tb-BSF buffer (90 mM NaCl, 5 mM KCl, 0.15 mM CaCl, 50 mM HEPES, pH 7). The cells were then resuspended in 150 μl of 1× Tb-BSF buffer, mixed with 10 μg of precipitated DNA and then transferred into a transfection cuvette. The cells were then electroporated using the Amaxa Nucleofector II and programme Z-001. Cells were recovered in 25 ml of HMI-9 supplemented with 20% FBS v/v for at least 5 h, after which they were selected with the appropriate drugs (Blasticidin, 2.5 μg/ml;

Phleomycin 0.5 μg/ml). The primers and plasmids used are listed in Supplementary Tables 2, 3, 4 and 5.

**Statistical analyses and reproducibility**. Graphical and statistical analyses were carried out in GraphPad Prism version 9 (GraphPad Software, La Jolla, California, USA, https://www.graphpad.com) by two-way repeated-measures ANOVA test followed by Tukey or Šídák post hoc analysis. For individual experiments, *n* values are included in the Figure legend; graphs provide mean values ± SEM. *P* values of less than 0.05 were considered statistically significant. For infection profiles, western blots and cytological studies (e.g. PAD assays) experiments were initially tested as pilot assays; where individual experiments are presented in the manuscript figures these are consistent with these pilot assays.

**Reporting summary**. Further information on research design is available in the Nature Research Reporting Summary linked to this article.

## Data availability

The mass spectrometry proteomics data generated in this study have been deposited to the ProteomeXchange Consortium via the PRIDE[36] partner repository (https://www.ebi.ac.uk/pride/archive) with the dataset identifier PXD032101. The data comprising parasitaemia scores and gel images are provided with this paper in the accompanying source data files and supplementary information. All biological reagents are available upon request subject to availability or regulatory approval Source data are provided with this paper.

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

## Acknowledgements
Mass spectrometry was carried out on equipment awarded to the Centre for Immunity, Infection and Evolution (Wellcome Trust award 095831/Z/11/Z). The proteomics analyses were carried out by the EdinOmics research facility at the University of Edinburgh and we particularly acknowledge the assistance of Lisa Imrie. We also thank Dr. Nisha Philip (University of Edinburgh) and Professor Jay Bangs (Buffalo University, New York) for invaluable comments and input. Images in Fig. 8 were created with Biorender.com. Work in Keith Matthews laboratory is funded by the Wellcome Trust (103740/Z/14/Z, 221717/Z/20/Z). M.D.T. was supported by the Darwin Trust of Edinburgh. This research was funded in whole, or in part, by the Wellcome Trust [103740/Z/14/Z, 221717/Z/20/Z]. For the purpose of open access, the author has applied a CC BY public copyright licence to any Author Accepted Manuscript version arising from this submission.

## Author contributions
The conceptualisation and conduct of the experiments, in addition to the data analysis, were carried out by MT, FR and KM. Supervision of the work was performed by FR and KM. The manuscript was written and revised by MT, FR and KM. Project administration and funding acquisition was carried out by KM.

## Competing interests
We declare that the authors have no competing interests as defined by Nature Portfolio, or other interests that might be perceived to influence the results and/or discussion reported in this paper.
