## [Peer Review File · Nature Communications]

First round comments -

Reviewer #1 (Remarks to the Author):

The Matthew's lab is a leader in deciphering the trypanosome quorum sensing (QS) response. In particular, the discovery that short peptides could promote trypanosome differentiation raised the possibility that parasite-derived peptidases could generate the extracellular signal for QS. This led to the current study where they analyzed the peptidases released by trypanosomes and evaluated their ability to contribute to the generation of the QS response. This is a logical extension of previous results and provides some new functional clues to how QS works. However, there are some major concerns.

Major points:

1. One of the major points of the manuscript is that two of the peptidases identified dominate generation of the QS signal. This might be the case, but it is unclear to this reviewer how the generation of the QS signal is quantitated. Is there a scale the authors have established and can be applied to different experiments. In other words what does dominance in the generation of the QS signal mean?
2. Overall the data presentation is rather unconventional and lacks statistical analysis. In numerous figures replicates are presented as separate data points, which makes it very difficult, or even impossible, to fully appreciate the data. Replicates need to be combined and presented with error bars. Even though it is stated in Fig. legends that three replicates were performed, often only two experiments are visible. It seems very unlikely that independent experiments result in identical data points.
3. There appears to be some discrepancies in the apparent stringent inducible regulation of expression (Fig. 2B versus Supplementary Fig 2). There is some leakage of expression in Fig. 2B in Tb927.6.400, Tb927.8.7020, Tb927.3.2090, Tb927 .11.3570, and Tb927 .11.2500, as well as in Tb927.3.4750 in Supplementary Fig 2. The statement on line 198 needs to be changed and an explanation of this variation needs to be provided. In addition, the effects on cell growth are described with vague terms (substantially reduced growth upon induction, reduced the growth of the cells somewhat, had minimal effects). A more quantitative description would be helpful.
4. Although deleting two peptidases (Fig. 6) generated a parasitemia of enhanced virulence, the single and double peptidase deletion lines were not significantly different from each other with respect to the PAD1 marker. How do the authors explain this result?
5. The add back of Tb927.11.2500 in null mutants did not result in establishing infections in mice or restoration of stumpy formation and thus, questioning the direct involvement of Tb927.11.2500. The authors provide no explanation or discussion for this result, which leaves this reviewer somewhat unsatisfied.
6. The data shown in Fig. 3B (Western blot) is of rather poor quality with numerous degradation products and thus, it is impossible to judge the author's claim that "We observed no correlation between those peptidases that induced enhanced differentiation when induced (Figure 3A) and their relative expression or release as assayed by western blotting (Figure 3B)."
7. Prolyl oligopeptidase and pyroglutamyl peptidase have previously been identified and are released by bloodstream form parasites. Why were these two not detected as components of the secreted material in this study?

Minor comments:

1. The authors claim that the release of the identified peptidase is mediated via unconventional protein secretion. This might be the case, but they only showed this for one peptidase, so the

language has to be toned down.

2. Figure 4B with Tb927.8.7020 OE/ RBP7 KO does not show triplicates of the +Dox and the Figure is messed up.

3. Line 219, It is unacceptable to have data not shown: "the expression of the stumpy specific marker protein PAD1(Dean et al., 2009) was detectable (not shown)."

4. Line 238, RBP7 is present in two copies and thus deletion of this gene, as stated, is not very clear.

Reviewer #2 (Remarks to the Author):

- **Synopsis:** This group previously identified an oligopeptide transporter as contributing to QS-induced slender to stumpy formation in *T. brucei*. That finding led them to discover oligopeptides inducers of stumpy formation, and further to find that ectopic secretion of peptidase from *T. brucei* could induce stumpy formation in trans. Here they use rigorous proteomics and protein tagging to identify endogenous secreted peptidases of *T. brucei* and then systematically assess each of these for its role in stumpy formation during mouse infection. Using inducible over-expression and gene knockout, they define three peptidases that contribute to generation of the stumpy inducing activity, and two of these as major contributors. They employ genetic interference assays to demonstrate that the identified peptidases operate through the RBP7 QS pathway. Finally, the two main peptidases are shown to act in concert to give maximum stumpy induction response and release is found to be a mechanism other than classical ERGolgi, or Rab11-dependent EV/nanotube release, though the mechanism was not defined.

- **Critique:** This is an excellent piece of work, addressing a long-standing and important problem in *T. brucei* biology and substantially advancing our knowledge of mechanisms used by trypanosomes to undergo developmental regulation of life cycle stages. The work is thorough, rigorous and compelling, including independent tests of peptidase secretion and function, multiple replicates in vitro and subsequent assessment during mouse infection for activity, independent assessment of over/ectopic expression and gene KO for function. The results are very compelling and conclusions well-supported. I expect the work to be of very broad interest in the field. I only have a few minor comments for the authors to address:

1) Fig 1C is not described well in the text or fig legend, e.g. "fold change" vs what?? Please provide a more clear description of the questions being addressed.

2) For gene ... that showed a strong phenotype in vitro, should this gene not also be assessed during infection? I wouldn't make that a requirement for publication, but ask the authors to address. Maybe it's an important contributor to QS signaling?

3) Fig 3: labeling of some panels was out of alignment.

4) In some figures, the "12850" gene is labeled as "12580" (5 and 8 transposed). Please correct.

5) Fig 4C: the quantitation of PAD expression in RBP7KO with peptidase OE, - v + tet is hard to interpret without comparable analysis of PAD expression in the WT background. The authors state PAD1 expression observed in +Tet in WT background. It would be best if some data on that can be shown for comparison w Fig 4. The parasitemia curves in Fig 4 are compelling, e.g. vs those in Fig 3, but PAD1 expression data is hard to evaluate vs WT background. Please clarify.

6) Please clarify the term "overexpression". I realize the expression is inducible, but do the authors have independent test of whether this represents "over"expression vs WT levels?

7) Fig 6D: it appears that add back does not restore PAD1 expression, despite restoring some ability to slow growth during infection (fig 6B, parasitemia analysis). Please clarify.

8) Line 335: consider using "cytoplasmic" (in the cytoplasm as a cytological definition), vs "cytosolic" (soluble portion of the cytoplasm as a biochemical fractionation definition).

9) Regarding the unconventional protein secretion pathway for QS peptidases, have the authors

considered ciliary ectosomes? In some organisms ciliary ectosomes are enriched for proteases e.g. Long, H. et al. *Curr Biol* 26, 3327–3335 (2016). Trypanosomes have been shown to release extracellular vesicles that appear to be derived from the flagellum Szempruch, A. J. et al. *Cell* 164, 246–257 (2016). Since mechanisms of ciliary vesicle release are not well defined, this could be a route to consider for release of the peptidases identified.

Reviewed by: Kent Hill

Reviewer #3 (Remarks to the Author):

African trypanosomes that are growing as bloodstream forms, in either culture or mammalian hosts, enter stationary phase when they reach a certain density. The resulting parasites have changed gene expression and are much better able to survive in tsetse flies than cells that are in log phase. It has long been known that a quorum sensing mechanism can induce stumpy formation. The authors have previously shown that quorum sensing in vitro occurs via peptides that accumulate in the medium, generated by trypanosome-derived proteases. They already identified various players in the pathway, including the peptide receptor. In this paper the authors identify secreted proteins, and go on to find two secreted peptidases that play prominent and additive roles in generating the signal. One of them is shown to be secreted by an "unconventional" (non signal-peptide-mediated) mechanism. In general the paper is convincing, I do think that one experiment is missing (unless I overlooked it) and the full mass spectrometry results must be shown and deposited in a database. The remaining comments are minor.

Experiment:

To find out whether the quorum sensing pathway was involved the authors used lines lacking RBP7. I found this a rather odd choice since RBP7 is downstream of the YAK kinase in the pathway, and more importantly, its precise role/mechanism is (so far as I know) unknown. The authors should in addition directly test protease involvement in creating the inducing signal. This is classically done by taking supernatants from cultured parasites and assessing their effects on long slender cells at low density. In this case the double knockout supernatants should have reduced stumpy-inducing capability. I think this is an essential (and trivial) experiment.

Mass spectrometry:

All data must be deposited in a suitable database and the full list of all detected proteins, with LFQ, peptides, etc. and the statistics, must be presented as a supplementary table. Also Figure 1C is unclear. Which samples are to the left, and which to the right on each of the two graphs? These must be labelled, at present I can't tell, at all, what is being compared with what.

Changes/additions to text:

- a) The authors should very briefly state (with references) that there is evidence that stumpy formation can happen via stimuli other than quorum sensing, and that the full pathway is not absolutely required for differentiation into the procyclic form. Neither detracts from the current paper in the least.
- b) The locations of GFP-tagged versions of these peptidases inside procyclic-form trypanosomes have previously been determined by the TrypTag project. These results must be described and cited. Of course they are high-throughput so needed verification, but they also help the authors since in many cases results for both N- and C-terminally tagged protein are available. Relevant to this - Paragraph beginning 182 - where was the tag for over-expression? What is the location of these when C-terminally tagged (maybe from tryp-tag results)?
- c) Please supply a Table with details of all of the plasmids (or PCR-products, if used).

Other (mostly typos/phrasing)

Two peptidases could not be inducibly over-expressed. and one (Tb927.8.8330) gave a growth defect and was not further analysed. Did the authors at least check PAD1 expression in vitro after Tb927.8.8330 over-expression?

91 - a caveat that these studies were RNAi should be added, since RNAi cannot demonstrate that a gene product is dispensable.

lefthand should be left-hand, right-hand also hyphenated..

line 228 - I think this should be "of their higher expression".

Line 190 - There is a typo here, Tb927.10.12660 is PUF2

Line 383 : typo I think: "null mutants affected overall" should be "null mutations" or "mutants showed reduced differentiation". The mutant is the organism itself so it can't "affect" something.

Line 406: reference missing!!

In several places (Figures, Figure legends) the antigenic type name "AnTat 1.1" appears. Do the authors know which VSG is expressed in all these cases? If not the label should be changed to "EATRO1125".

Figure 7: Title is too general. It should be "Oligopeptidase B" not "released peptidases".

REVIEWER COMMENTS

Reviewer #1 (Remarks to the Author):

The Matthew's lab is a leader in deciphering the trypanosome quorum sensing (QS) response. In particular, the discovery that short peptides could promote trypanosome differentiation raised the possibility that parasite-derived peptidases could generate the extracellular signal for QS. This led to the current study where they analyzed the peptidases released by trypanosomes and evaluated their ability to contribute to the generation of the QS response. This is a logical extension of previous results and provides some new functional clues to how QS works. However, there are some major concerns.

Major points:

1. One of the major points of the manuscript is that two of the peptidases identified dominate generation of the QS signal. This might be the case, but it is unclear to this reviewer how the generation of the QS signal is quantitated. Is there a scale the authors have established and can be applied to different experiments. In other words what does dominance in the generation of the QS signal mean?

Our evidence for the dominance of the peptidases is based on compounded evidence through a systematic series of studies:

- The ability of respective peptidases to drive the QS response was demonstrated by ectopic overexpression and gene knockout, with effects determined by the quantitative analysis of parasite numbers and development to stumpy forms where informative (e.g. altered parasitaemia due to anticipated accelerated or delayed differentiation).
- Critically, all effects were determined in vivo allowing a comparison of the contribution of individual peptidases to quorum sensing in a physiological setting, rather than in culture. In each case, the parasitaemias were quantitated and statistically assessed (the error bars and P values are now included in the revised manuscript as requested) and those demonstrating significant effects pursued.
- After systematic over-expression of each peptidase, an enhanced QS response was detected by parasites becoming stumpy at low parasitaemia. Three peptidases generated accelerated differentiation by ectopic expression and we confirmed that the response was dependent on quorum sensing rather than other factors by analysing the same overexpression in a QS-defective cell line (RBP7A/B null). This demonstrated the peptidase expression promoted differentiation through the QS pathway.
- Thereafter, null mutants were generated for the three QS active peptidases and two of them showed reduced differentiation in vivo, again quantitated by parasite number, developmental marker expression and statistical validation.
- Further, the combined null mutation of both of these genes further reduced differentiation (and so enhanced virulence), the effect being confirmed by the add-back controls.

In combination, two peptidases were determined to dominate the response since only these molecules of the 12 analysed reduced differentiation when deleted and enhanced differentiation when overexpressed. Of course, the combined action of other peptidases

could provide a further minor contribution to the QS signal as we discuss in the manuscript and represented in Figure 8, but these would be individually less effective than the dominant peptidases Oligopeptidase B and metalloprotease 1.

2. Overall the data presentation is rather unconventional and lacks statistical analysis. In numerous figures replicates are presented as separate data points, which makes it very difficult, or even impossible, to fully appreciate the data. Replicates need to be combined and presented with error bars. Even though it is stated in Fig. legends that three replicates were performed, often only two experiments are visible. It seems very unlikely that independent experiments result in identical data points.

We showed individual datapoints for each mouse infection since we thought this was the fairest way to represent the inherent variability that normally occurs between individual parasitaemias in rodents. However, as requested we have now represented the data as mean \pm SEM if this is the preferred representation. In some cases, the variation between replicates was indeed too small to visualise on the original graphs, but with the data now represented as mean and error, this is no longer relevant for the data presentation. In other cases, the variability characteristic of in vivo infections generates larger error bars when amalgamated, despite individual parasite infection profiles making the outcome clear. Nonetheless, we pursued only robust effects and all raw data is provided in the Supplementary data file as mandated by the journal and so is available for inspection and further analysis by readers.

3. There appears to be some discrepancies in the apparent stringent inducible regulation of expression (Fig. 2B versus Supplementary Fig 2). There is some leakage of expression in Fig. 2B in Tb927.6.400, Tb927.8.7020, Tb927.3.2090, Tb927 .11.3570, and Tb927 .11.2500, as well as in Tb927.3.4750 in Supplementary Fig 2. The statement on line 198 needs to be changed and an explanation of this variation needs to be provided. In addition, the effects on cell growth are described with vague terms (substantially reduced growth upon induction, reduced the growth of the cells somewhat, had minimal effects). A more quantitative description would be helpful.

The ectopic expression of the peptidases was very stringent, but of course there is always an element of leakiness in inducible systems. This can be detected in the cell fractionation (pellet / supernatant) assays since these concentrate the protein samples of each fraction, this not being routinely seen in the whole cell extracts in Supplementary figure 2B. In terms of the phenotypes observed there was no clear consequence of this inevitable slight leakiness- three peptidases generated robust inducible phenotypes in vivo, no others did.

For the in vitro growth analyses, we agree with the referee that some more quantitative descriptors could help categorisation. Hence, we have defined a growth effect of <10% as 'no effect', 10-50% as 'moderate' and >50% as strong. This is now annotated in the text (line 199-206).

4. Although deleting two peptidases (Fig. 6) generated a parasitemia of enhanced virulence, the single and double peptidase deletion lines were not significantly different from each other with respect to the PAD1 marker. How do the authors explain this result?

This is the expected outcome. With one peptidase deleted there is significantly reduced differentiation that is fully restored with add back. However, when the genes for the two peptidases are deleted there is further reduction in differentiation which is represented as elevated parasitaemia. The add back simply restores the parasites to the level of reduced differentiation seen with the single knockout which is more virulent and expresses less PAD1 than wild type. PAD1 expression typically lags growth arrest in the normal QS response

pathway and so detectable differences between the single and double knockout lines would not be expected: both are defective in their QS signal generation although the effect is more extreme when both peptidases are absent. Restoration to wild type differentiation levels would not be expected, and was not observed and this is now further clarified on lines 322-323. As an aside, note that the add-back in each case involved reintegration into the endogenous locus rather than introduction of an ectopic copy elsewhere in the genome which could be potentially over expressed.

5. The add back of Tb927.11.2500 in null mutants did not result in establishing infections in mice or restoration of stumpy formation and thus, questioning the direct involvement of Tb927.11.2500. The authors provide no explanation or discussion for this result, which leaves this reviewer somewhat unsatisfied.

We do not have an explanation for this interesting result, which we would like to explore further although it is beyond the scope of the current manuscript. We wondered, for example, if the Tb927.11.2500 add-back might be less able to disseminate in tissues and so could not establish infections when inoculated interperitoneally (our normal infection route). However, this was not obviously the case – we found infection intravenously also did not allow infections to establish. Importantly, this phenotype was reproducible for two independently derived clones. Hence, the inability to infect mice represents a feature of the peptidase add-back, which is why we were cautious in our text when describing the consequences on differentiation of add back for this peptidase. We have also expanded our description of the phenotype in the revised manuscript (line 295-298).

6. The data shown in Fig. 3B (Western blot) is of rather poor quality with numerous degradation products and thus, it is impossible to judge the author's claim that "We observed no correlation between those peptidases that induced enhanced differentiation when induced (Figure 3A) and their relative expression or release as assayed by western blotting (Figure 3B)."

We disagree with this point and believe it is clear that the strong differences in phenotype observed for parasites analysed in vivo are not attributable to the trivial explanation that there were gross differences in the level of expressed peptidase in each cell line. The western blots represent cell pellet and purified supernatant from the respective cell lines where the relative levels of each peptidases can be compared – the cells lines were grown and analysed, and protein extracted, in parallel. Although some protein fragments are detected on the cell supernatant this is not unusual for such isolations. But clearly those peptidases generating the strongest phenotypes were not simply those that were expressed at the highest level.

7. Prolyl oligopeptidase and pyroglutamyl peptidase have previously been identified and are released by bloodstream form parasites. Why were these two not detected as components of the secreted material in this study?

Prolyl oligopeptidase was detected in our mass spectrometry analysis, but below the cut off threshold that we applied for selection (Supplementary datafile 1, 2). PGP was not detected, perhaps being released in insufficient quantities for detection. In fact, we have evaluated the contribution of POP by gene knockout and this does not reduce parasite differentiation, despite its ectopic overexpression accelerating development (Rojas et al., Cell, 2019). Hence it does not appear to be a dominant contributor to generation of the QS signal unlike OPB and MCP1. This is further supported by previous studies that showed that when OPB is deleted, the activity of POP generated by parasites increases (PMID: 22123425) . In this situation, we would not have expected to see a reduction in differentiation in our OPB null mutants if POP could compensate as a significant contributor to production of the QS signal. Our interpretation is that POP and PGP can promote differentiation when ectopic

overexpressed but they do not importantly contribute to the QS signal individually. Instead, as we demonstrate by ectopic expression, gene knockout and combinatorial gene knockout, OPB and MCP1 dominate.

Minor comments:

1. The authors claim that the release of the identified peptidase is mediated via unconventional protein secretion. This might be the case, but they only showed this for one peptidase, so the language has to be toned down.

We have been explicit that this was tested directly for only OPB in the manuscript (including the Abstract). Although we think likely it will be relevant for all the peptidases studies, we have checked the manuscript to ensure we do not overclaim on this point.

2. Figure 4B with Tb927.8.7020 OE/ RBP7 KO does not show triplicates of the +Dox and the Figure is messed up.

The third replicate was not included for this one western blot because the parasites were not successfully purified at the end of the infection after a failure with the DEAE resin. However, this does not affect the overall conclusions of the assays since two other replicates were successfully purified and analysed. We have added a note to the Figure 4B legend to reflect the missing sample.

The labelling on the figures has been corrected- thank you for pointing out this mistake.

3. Line 219, It is unacceptable to have data not shown: "the expression of the stumpy specific marker protein PAD1(Dean et al., 2009) was detectable (not shown)."

We have now included the quantitation of the PAD1 expression. For clarity and space constraints on the figures we had not included it, but have now added the data as a further supplementary figure (Supplementary Figure 7a). PAD1 levels were similar between induced and uninduced parasites at day 6 despite the much lower parasitaemia of the induced parasite lines.

4. Line 238, RBP7 is present in two copies and thus deletion of this gene, as stated, is not very clear.

This is correct- RBP7A and RBP7B are tandemly linked as two genes. The RBP7 null mutant deletes both copies, as previously described (MacDonald et al 2017). We have clarified this in the revised manuscript.

Reviewer #2 (Remarks to the Author):

- Synopsis: This group previously identified an oligopeptide transporter as contributing to QS-induced slender to stumpy formation in *T. brucei*. That finding led them to discover oligopeptides inducers of stumpy formation, and further to find that ectopic secretion of peptidase from *T. brucei* could induce stumpy formation in trans. Here they use rigorous proteomics and protein tagging to identify endogenous secreted peptidases of *T. brucei* and then systematically assess each of these for its role in stumpy formation during mouse infection. Using inducible over-expression and gene knockout, they define three peptidases that contribute to generation of the stumpy inducing activity, and two of these as major

contributors. They employ genetic interference assays to demonstrate that the identified peptidases operate through the RBP7 QS pathway. Finally, the two main peptidases are shown to act in concert to give maximum stumpy induction response and release is found to be a mechanism other than classical ERGolgi, or Rab11-dependent EV/nanotube release, though the mechanism was not defined.

• Critique: This is an excellent piece of work, addressing a long-standing and important problem in *T. brucei* biology and substantially advancing our knowledge of mechanisms used by trypanosomes to undergo developmental regulation of life cycle stages. The work is thorough, rigorous and compelling, including independent tests of peptidase secretion and function, multiple replicates in vitro and subsequent assessment during mouse infection for activity, independent assessment of over/ectopic expression and gene KO for function. The results are very compelling and conclusions well-supported. I expect the work to be of very broad interest in the field. I only have a few minor comments for the authors to address:
1) Fig 1C is not described well in the text or fig legend, e.g. "fold change" vs what?? Please provide a more clear description of the questions being addressed.

We have clarified this in the revised Figure 1C legend as well as the associated manuscript text (line 141-152)

2) For gene ... that showed a strong phenotype in vitro, should this gene not also be assessed during infection? I wouldn't make that a requirement for publication, but ask the authors to address. Maybe it's an important contributor to QS signaling?

Ectopic expression of this gene (Tb927.8.8330) resulted in the parasites dying very quickly upon induction (and not simply arresting in their growth, as with other peptidases). Hence, we did not assess it in vivo since the death of the parasites would not generate an informative outcome that would allow justification of the animal usage. This was detailed in the manuscript but has now been made more explicit.

3) Fig 3: labelling of some panels was out of alignment.

Corrected- our apologies we had missed this

4) In some figures, the "12850" gene is labelled as "12580" (5 and 8 transposed). Please correct.

Corrected- our apologies we had missed this as well!

5) Fig 4C: the quantitation of PAD expression in RBP7KO with peptidase OE, - v + tet is hard to interpret without comparable analysis of PAD expression in the WT background. The authors state PAD1 expression observed in +Tet in WT background. It would be best if some data on that can be shown for comparison w Fig 4. The parasitaemia curves in Fig 4 are compelling, e.g. vs those in Fig 3, but PAD1 expression data is hard to evaluate vs WT background. Please clarify.

Our aim was to compare induced and uninduced expression of the peptidases in the RBP7A/B null background and the included blots (panel B) and cell based scoring (Panel C) fulfil this. As the referee highlights, the parasitaemias make clear the experiment outcome. However, it is fair to highlight that a stumpy control would have been helpful to compare to the cells with differentiation capacity analysed in the experiment. Unfortunately, we did not do this at the time (the parasitaemias and other parameters made the outcome clear) and we do not feel it is justified to fully repeat the experiment with associated animal usage. However, we have re-run one of the western blots (OPB ectopic expression) with an

additional stumpy sample and the result is added as a supplementary figure (Supplementary figure 7C). This shows much less signal from the peptidase expressing lines (regardless of induction) in the RBP7A/B KO line compared to bona fide stumpy forms supporting involvement of the functional QS signalling pathway in generating the peptidase generated developmental response

6) Please clarify the term "overexpression". I realize the expression is inducible, but do the authors have independent test of whether this represents "overexpression" vs WT levels?

We have determined this for OPB and MCP1 ectopic expression using quantitative western blotting. The ectopic expression results in a 5.6 and 6.3 fold increase in the peptidase levels respectively. This is now included in the manuscript text (line 190) and included as supplementary figure 7B.

7) Fig 6D: it appears that add back does not restore PAD1 expression, despite restoring some ability to slow growth during infection (fig 6B, parasitemia analysis). Please clarify.

As highlighted in our response to referee 1 this is the expected outcome. With one peptidase deleted there is significantly reduced differentiation, that is fully restored with add back. However, when two peptidases are deleted, there is furtherer reduction in differentiation, but the add back simply restores the parasites to the level of reduced differentiation seen with the single knockout. Restoration to wild type differentiation levels would not be expected, and was not observed. As an aside, note that the add-back in each case involved reintegration into the endogenous locus rather than introduction of an ectopic copy elsewhere which could be potentially over expressed.

8) Line 335: consider using "cytoplasmic" (in the cytoplasm as a cytological definition), vs "cytosolic" (soluble portion of the cytoplasm as a biochemical fractionation definition).

Now amended thank you.

9) Regarding the unconventional protein secretion pathway for QS peptidases, have the authors considered ciliary ectosomes? In some organisms ciliary ectosomes are enriched for proteases e.g. Long, H. et al. *Curr Biol* 26, 3327–3335 (2016). Trypanosomes have been shown to release extracellular vesicles that appear to be derived from the flagellum Szempruch, A. J. et al. *Cell* 164, 246–257 (2016). Since mechanisms of ciliary vesicle release are not well defined, this could be a route to consider for release of the peptidases identified.

The release of EVs by trypanosomes has been observed and can be manipulated by RNAi against Rab 11 that directs endosomal recycling (Umaer et al., 2018; PMID: 29582527). Therefore, we applied knock down of Rab11 in a cell line with epitope tagged OPB1- this generating the expected rapid cessation of growth within 12-24h, and then cell death confirming the previously observed phenotype. However, examining the first few hours of induction did not reveal a difference in the extent of OPB release into the medium- suggesting EV release is not a contributor. We included this result in our manuscript in Figure 7C. We had referred to the outcome in the Discussion of the manuscript (line 413-424) which we hope makes things explicit. The ciliary ectosome type release occurs it must be via a distinct mechanism to EV release involving Rab11, which we refer to as 'other uncharacterised UPS routes' in the manuscript.

Reviewed by: Kent Hill

Reviewer #3 (Remarks to the Author):

African trypanosomes that are growing as bloodstream forms, in either culture or mammalian hosts, enter stationary phase when they reach a certain density. The resulting parasites have changed gene expression and are much better able to survive in tsetse flies than cells that are in log phase. It has long been known that a quorum sensing mechanism can induce stumpy formation. The authors have previously shown that quorum sensing in vitro occurs via peptides that accumulate in the medium, generated by trypanosome-derived proteases. They already identified various players in the pathway, including the peptide receptor.

In this paper the authors identify secreted proteins, and go on to find two secreted peptidases that play prominent and additive roles in generating the signal. One of them is shown to be secreted by an "unconventional" (non signal-peptide-mediated) mechanism. In general the paper is convincing, I do think that one experiment is missing (unless I overlooked it) and the full mass spectrometry results must be shown and deposited in a database. The remaining comments are minor.

Experiment:

To find out whether the quorum sensing pathway was involved the authors used lines lacking RBP7. I found this a rather odd choice since RBP7 is downstream of the YAK kinase in the pathway, and more importantly, its precise role/mechanism is (so far as I know) unknown.

We chose to use the RBP7A/B knockout since this is known to ablate the ability of trypanosomes to undergo quorum sensing- the question being addressed. Also, RBP7- a predicted RNA regulator- has previously been analysed by gene knockout, RNAi and global RNAseq analysis and its positioning in the QS signalling hierarchy has been examined (Mony et al., Nature 2014, PMID: 24336212 ; McDonald et al., PLOS Pathogens 2018, PMID: 29940034). But, of course, we could have used any component of the QS signalling pathway to address the question with the same outcome. We asked 'Do the peptidase expressions drive growth or developmental effects when the QS signalling pathways is disrupted?' and the answer was 'no'.

The authors should in addition directly test protease involvement in creating the inducing signal. This is classically done by taking supernatants from cultured parasites and assessing their effects on long slender cells at low density. In this case the double knockout supernatants should have reduced stumpy-inducing capability. I think this is an essential (and trivial) experiment.

Trypanosomes do not undergo quorum sensing in liquid culture media and so the experiment described is not straightforward and requires the generation of conditioned medium and exposure of parasites to Basement membrane extract, that supports differentiation in culture. Furthermore, we think it is unlikely to provide a definitive outcome because peptidases released in culture have different stability, turnover and substrate availability compared to the situation in vivo. In our manuscript we were careful to evaluate effects in their physiological setting in vivo, where the contribution of the peptidase was clear. In vitro, other peptidases providing additional minor contributions to the QS signal generation can accumulate and confound the proposed experiment. Nonetheless for the referee's interest we have attempted the suggested experiment using the supernatant from wild type or double peptidase KO lines as conditioned medium for wild type cells. We saw a small reduction in developmental induction with the double knock out line and reduced PAD expression, consistent with the expected outcome. However, due to the unphysiological nature of the experiment we do not think it is particularly informative and prefer not to include it in the manuscript.

Effect of conditioned medium from wild type or double peptidase knockout lines on the growth and differentiation of wild type cells. The left panel shows the growth of the parasites, with slightly higher growth when using conditioned medium from the double KO line. Conversely these cells showed reduced expression of the PAD1 stumpy marker.

Mass spectrometry:

All data must be deposited in a suitable database and the full list of all detected proteins, with LFQ, peptides, etc. and the statistics, must be presented as a supplementary table. Also Figure 1C is unclear. Which samples are to the left, and which to the right on each of the two graphs? These must be labelled, at present I can't tell, at all, what is being compared with what.

The raw data etc. has now been uploaded to the PRIDE proteomics database (dataset identifiers PXD032101 and 10.6019/PXD032101) and the relevant accession details are provided in the manuscript (line 646-650). The legend to Figure 1C has been expanded.

The reviewer access code is copied below

Username: reviewer_pxd032101@ebi.ac.uk
Password: pyANGtFJ

Changes/additions to text:

a) The authors should very briefly state (with references) that there is evidence that stumpy formation can happen via stimuli other than quorum sensing, and that the full pathway is not absolutely required for differentiation into the procyclic form. Neither detracts from the current paper in the least.

Stumpy formation in vitro has been induced artificially using engineered overexpression of a second VSG gene copy but the physiological relevance of this in vivo is unclear and not relevant to the discussion of quorum sensing. No other factors have been identified that have been shown to promote physiological stumpy formation to our knowledge. The assessment of what constitutes a functional stumpy form (and so required for differentiation to procyclic forms) is currently debated but we referred in the manuscript discussion to the relevant manuscripts discussing this issue in relation to the transmissibility of small numbers of parasites in livestock (lines 425-449).

b) The locations of GFP-tagged versions of these peptidases inside procyclic-form trypanosomes have previously been determined by the TrypTag project. These results must be described and cited. Of course they are high-thruput so needed verification, but they also help the authors since in many cases results for both N- and C-terminally tagged protein are available. Relevant to this - Paragraph beginning 182 - where was the tag for over-expression? What is the location of these when C-terminally tagged (maybe from tryptag results)?

c) Please supply a Table with details of all of the plasmids (or PCR-products, if used).

We apologise that we were not explicit about the tag location when first introduced. The tag was introduced at the N terminus for both the endogenously tagged and ectopically expressed copies. This was detailed on line 159 of the submission when the endogenously tagged copies were described but not referred again for ectopic expression. We have now inserted this more clearly into the results section. We have assessed the TrypTag dataset and the peptidases we analysed were cytosolic when C-terminally tagged. This is tabulated below.

Supplementary Tables have now been included for the details of plasmids and primers as requested (Supplementary Tables 4, 5 and 6).

Gene IDs	Protein description	TrypTag Protein Localisation	
		N-terminus tag	C-terminus tag
Tb927.11.2500	Metalloprotease 1	Cytoplasm	cytoplasm flagellar cytoplasm Basal body
Tb927.8.8330	Calpain	cytoplasm (reticulated, weak)	pellicular membrane (50%)
Tb927.3.2090	Aminopeptidase P1	flagellar cytoplasm cytoplasm	cytoplasm flagellar cytoplasm
Tb927.8.7020	Peptidase 1	cytoplasm flagellar cytoplasm	cytoplasm flagellar cytoplasm nucleoplasm (weak)
Tb927.11.6590	Aminopeptidase 2		cytoplasm
Tb927.6.400	Peptidase M20/M25/M40		cytoplasm flagellar cytoplasm
Tb927.11.3570	Aminopeptidase 1		cytoplasm flagellar cytoplasm
Tb927.3.3410	Aspartyl aminopeptidase	cytoplasm (patchy)	cytoplasm (points)
Tb927.10.12260	Cytosolic nonspecific dipeptidase	cytoplasm flagellar cytoplasm (weak)	cytoplasm (75%) flagellar cytoplasm (75%)
Tb927.11.12850	Oligopeptidase B	cytoplasm (reticulated)	cytoplasm flagellar cytoplasm
Tb927.3.4750	Aminopeptidase 3		flagellar cytoplasm (strong) nucleoplasm cytoplasm
Tb927.1.2100	Calpain-like cysteine peptidase	paraflagellar rod flagellar tip (strong, cell cycle dependent)	

Other (mostly typos/phrasing)

Two peptidases could not be inducibly over-expressed. and one (Tb927.8.8330) gave a growth defect and was not further analysed. Did the authors at least check PAD1 expression in vitro after Tb927.8.8330) over-expression?

As detailed for referee 2, induction for the ectopic expression of this gene caused rapid cell death. PAD1 staining was not possible.

91 - a caveat that these studies were RNAi should be added, since RNAi cannot demonstrate that a gene product is dispensible.

We have now added this as requested. Thank you for the suggestion.

lefthand should be left-hand, right-hand also hyphenated..

Changed

line 228 - I think this should be "of their higher expression".

Changed as suggested- thanks for highlighting this.

Line 190 - There is a typo here, Tb927.10.12660 is PUF2

Changed- apologies for the mistake.

Line 383 : typo I think: "null mutants affected overall" should be "null mutations" or "mutants showed reduced differentiation". The mutant is the organism itself so it can't "affect" something.

Thank you- we have rephrased the sentence.

Line 406: reference missing!!

In fact, several references were missing in the submitted manuscript due to an EndNote problem. We think we have now fixed it.

In several places (Figures, Figure legends) the antigenic type name "AnTat 1.1" appears. Do the authors know which VSG is expressed in all these cases? If not the label should be changed to "EATRO1125".

The nomenclature of the parental cell lines 'AnTat1.1 J1339', or 'AnTat 1.1. 90:13' reflects their naming in the source references where they were first described. Their designation as EATRO 1125 is specific in the Materials and Methods as is their subdesignation according to the source papers for the lines.

Figure 7: Title is too general. It should be "Oligopepidase B" not "released peptidases".

Amended.

Second round comments -

Reviewer #1 (Remarks to the Author):

The authors have satisfactorily addressed my comments and questions.

Reviewer #2 (Remarks to the Author):

The authors have satisfactorily addressed all my concerns and the revised manuscript is strong and compelling.

Reviewer #3 (Remarks to the Author):

I had mainly minor comments (apart from the data availability, which was major), and they have all been satisfactorily addressed.

REVIEWERS' COMMENTS

Reviewer #1 (Remarks to the Author):

The authors have satisfactorily addressed my comments and questions.

Reviewer #2 (Remarks to the Author):

The authors have satisfactorily addressed all my concerns and the revised manuscript is strong and compelling.

Reviewer #3 (Remarks to the Author):

I had mainly minor comments (apart from the data availability, which was major), and they have all been satisfactorily addressed.

The referees had nothing for us to address.